# The Potential of Senescence as a Target for Developing Anticancer Therapy

**DOI:** 10.3390/ijms24043436

**Published:** 2023-02-08

**Authors:** Hyein Jo, Kyeonghee Shim, Dooil Jeoung

**Affiliations:** Department of Biochemistry, College of Natural Sciences, Kangwon National University, Chuncheon 24341, Republic of Korea

**Keywords:** senescence, cancer, autophagy, histone deacetylases, microRNA, senolytics, senostatics

## Abstract

Senescence occurs in response to various stimuli. Senescence has attracted attention because of its potential use in anticancer therapy as it plays a tumor-suppressive role. It also promotes tumorigeneses and therapeutic resistance. Since senescence can induce therapeutic resistance, targeting senescence may help to overcome therapeutic resistance. This review provides the mechanisms of senescence induction and the roles of the senescence-associated secretory phenotype (SASP) in various life processes, including therapeutic resistance and tumorigenesis. The SASP exerts pro-tumorigenic or antitumorigenic effects in a context-dependent manner. This review also discusses the roles of autophagy, histone deacetylases (HDACs), and microRNAs in senescence. Many reports have suggested that targeting HDACs or miRNAs could induce senescence, which, in turn, could enhance the effects of current anticancer drugs. This review presents the view that senescence induction is a powerful method of inhibiting cancer cell proliferation.

## 1. Characteristics of Senescence

Cellular senescence is a state of permanent growth arrest and is thought to be an essential anticarcinogenic barrier in normal cells. Cellular senescence protects tissues from developing cancer by arresting the cell cycle [1,2,3,4,5,6,7,8]. Senescence can also create an inflammatory microenvironment favoring the initiation and progression of various age-related diseases including cancer [9]. Thus, senescence affects tumorigenesis in a context-dependent manner.

Senescent cells (SnCs) are enlarged and flattened and display increased cytoplasmic granularity. These cells remain viable and metabolically active and are different from quiescent cells, which are in a temporary state of growth arrest [10]. SnCs induced by different stimuli share the common characteristics of stable growth arrest, anti-apoptosis, persistent DNA damage response (DDR) signaling, changes in chromatin structures (senescence-associated heterochromatin foci), decreased nuclear lamin-B1 levels, and an increased expression of cyclin-dependent kinase (CDK) inhibitors (p16INK4a, p21Cip1/Waf1) [11,12,13], p53 [14], and senescence-associated β-galactosidase activity (SA-β-gal) [15,16]. Senescence is also characterized by changes in mitochondria dynamics, structure, and function [17]. Impaired mitochondrial dynamics lead to senescence [17]. Figure 1 shows the general characteristics of senescent cells. Senescent cells are unique in that they eventually stop multiplying but do not die off when they should. Senescent cells can remain active and spread senescence. Figure 1 also shows that senescent cells can spread senescence in paracrine and juxtacrine fashions.

## 2. Factors Inducing Senescence

Radiation has been shown to induced senescence [18,19,20,21,22]. Radiation induced senescence by increasing the expression of STAT3 and autophagic flux, such as Beclin1 and LC-3 II in lung adenocarcinoma cells [18]. This implies a close relationship between senescence and autophagy. Senescence occurs in response to chemotherapy, known as therapy-induced senescence (TIS) [14,16]. Low doses of chemotherapy trigger senescence, while higher doses trigger apoptosis. Paclitaxel (PTX)-induced senescence involved the increased expression of p53 in bladder cancer cells [14]. The accumulation of cryptochrome 1 (CRY1) in cisplatin-resistant cells prevented PTX-induced senescence by promoting p53 degradation [14]. Doxorubicin-induced senescence in cardiac fibroblasts was accompanied by increases in p53 and SA-β-gal activity in an IL-1β-dependent manner [16]. Etoposide, an inhibitor of topoisomerase, induced senescence by increasing the expression of p53 and p21 via DNA protein kinase (DNA-PK)-checkpoint kinase 2 (Chk2 pathway) in adrenocortical carcinoma cells [23].

The combination of a mouse double minute 2 (MDM2) antagonist with an inhibitor of the aurora kinase A (AURKA) activated p53, and induced growth arrest and the clearance of melanoma cells by antitumor leukocytes [24]. P53 displays diverse roles, including apoptosis and senescence. P53 activation in response to DNA damage causes growth arrest, allowing for DNA repair, or directs senescence or apoptosis [25]. For DNA repair, senescence, but not apoptosis, is induced by p53. P53 increases the expression of p21 and p16, which, in turn, causes cell cycle arrest to induce senescence [25]. These findings suggest a role of p53 in chemotherapy-induced senescence. It is probable that low doses of chemotherapy drugs cause senescence via p53, while higher doses of chemotherapy drugs cause apoptosis via p53. Platinum-based drugs, such as cisplatin, induce senescence by the upregulation of senescence-related genes (p53, p21, and p16) in seminoma cells [26]. Cisplatin induced DNA damage and increased ROS level [26].

The irreversible senescence is mediated by INK4a/Rb and p53 pathways, and the reversible senescent phenotype is mediated by p53. This suggests that the p53 pathway could be effectively harnessed as a therapeutic intervention to trigger senescence and ultimately suppress tumorigenesis.

Punicalagin (PUN), an active component from pomegranate, induced the senescence of human papillary thyroid carcinoma cells by increasing the expression of p21, interleukin-6 (IL-6), and IL-1β via nuclear factor NF-κB [27]. Chemotherapy-induced senescence involved the increased expression of anti-apoptotic proteins, such as BCL-2 and the induction of p53-mediated DNA damage response in T-cell lymphoma cell lines [28]. Senescent cells could persist without cell death owing to the upregulation of anti-apoptotic proteins such as BCL-2, and the downregulation of BAX and p53 in human dermal papilla cells [29]. This implies the role of senescence in anticancer drug resistance. Doxorubicin (DOX) induced endothelial cell (EC) senescence based on SA-β-gal activity, the upregulation of senescence markers (p53, p21, and cyclinD1), and the elevated expression of SASP [30].

Senescence can also be induced by inhibitors of CDK4/6 [31], polo and aurora kinases [32]. The CDK4/6 inhibitor (palbociclib) enhanced the anticancer effect of the I kappa B kinase (IKK)β inhibitor Bay 11-7082 in hepatocellular carcinoma [31]. The inhibition of aurora kinases induced polyploidy and the ATM/Chk2 DNA damage response, which mediated senescence and an NF-κB-related, senescence-associated secretory phenotype (SASP) [32]. This implies that AURK inhibitors can also induce senescence in a p53-independent way. The inhibition of HDAC2 or HDAC7 induced senescence by increasing the expression of p16 and p21 in dermal fibroblasts [33]. The downregulation of DNA methyl transferase 2 causes telomere shortening and senescence [34].

Senescence can also be induced by damage to mitochondria [35,36,37,38,39], which in turn, leads to the increased production of reactive oxygen species (ROS) [40] and DNA damage [41,42]. Mitochondrial dysfunction was shown to induce senescence via AMPK-p53 activation, and modified SASP without IL-1 in human fibroblasts [36]. Cathepsin F mediated the effect of DNA damage on the induction of senescence in human skin fibroblasts and keratinocytes [42]. Cellular senescence can also be induced by oxidative stress [43] and telomeric shortening [44]. Oxidative stress induced senescence by increasing SASP in human dental pulp cells [43]. Bortezomib, an anticancer drug, induced senescence by promoting telomeric shortening and increasing the expression of p53 and p21 in lung cancer cells [44]. Oxidative stress induced senescence and epithelial-to-mesenchymal transition (EMT) in amnionic epithelial cells (AEC) via p38MAPK activation [45]. Since therapeutic resistance eventually develops, it is probable that senescence induced by therapy may induce therapeutic resistance.

Figure 2A shows various factors inducing senescence and that senescence can promote both tumor suppression and tumor progression via SASP. Figure 2B shows therapy-induced senescence (TIS) and the mechanisms involved.

## 3. Oncogene-Induced Senescence

Oncogenes are known to induce senescence [45,46]. A combination of oncogenes (CTNNB1, TERT, and MYC) induced senescence in human fibroblasts and primary hepatocytes [47]. Transformed cells after senescence became more aggressive and display high invasion potential and anticancer drug resistance [46]. Continuous functional changes in senescent cells, and the permanent need to actively maintain senescence-supporting transcription puts the stability of cell cycle arrest at risk. Oncogene-induced senescence (OIS) can induce permanent growth arrest. OIS can also enable senescent cells to evade the potential toxicity of therapeutics, allowing for the eventual re-emergence or escape from senescence that could lead to disease recurrence. Senescence is potentially reversible through the inactivation of p53, p16(INK4A) and/or Rb, overexpression of Cdc2/cdk1, and the survival of cancer stem cells.

MYC is critical for the immune evasion of triple-negative breast cancer cells (TNBCs) [48]. TNBCs with high MYC are resistant to immune checkpoint inhibitor (ICI) therapy [48]. IL-6, a component of SASP, mediates the effect of MYC on somatic cell reprogramming (transformation) [49]. MYC is necessary for the SASP-induced somatic cell programming, such as the generation of induced pluripotent stem cells [49].

Oncogene-induced senescence (OIS) increased the production of ROS, which, in turn, caused DNA damage and DNA double-strand breaks (DSB) [5,50]. OIS increased cytoplasmic release of cytochrome c, the expression of BAX, and the cleavage of poly(ADP-ribose) polymerase (PARP) in human keratinocytes [50]. During OIS, Toll-like receptor 2 (TLR2) was shown to mediate senescence in vitro and in murine models [3]. TLR2 impairs early lung cancer progression by activating cell cycle arrest and the pro-inflammatory phenotype [3]. TLR2 promoted cell cycle arrest by increasing the expression of tumor suppressors (p53-p21CIP1, p16INK4a, and p15INK4b) by inducing the acute-phase serum amyloids A1 and A2 (A-SAAs) [51]. These reports suggest that OIS involves the initial activation of tumor-suppressive responses. It is also noteworthy that oncogenes that induce senescence can also promote tumor initiation by inducing immune evasion [48].

Oncogenic Ras transformation resulted in the upregulation of the cell cycle inhibitors p15Ink4b, p16Ink4a, and p19Arf; Rho-associated protein kinase (ROCK); SASP; and the downregulation of p-AKT in skin keratinocytes [52]. The upregulation of ROCK activity initiated a senescence response characterized by cell enlargement, growth inhibition, the upregulation of SA-β-gal activity, and the release of multiple pro-inflammatory factors, such as colony-stimulating factors (CSFs) and matrix metalloproteinase 9 (MMP-9) [52]. p53 and retinoblastoma (RB) act as master regulators of OIS [53,54]. The inactivation of p53 or p16 blocked the ability of oncogenic Ras to induce senescence in mouse fibroblasts [53]. Ras induced cellular senescence by promoting caspase-4 inflammasomes in a p53-dependent manner [54]. p53 was shown to be necessary for mitochondrial elongation in H-RAS-induced cellular senescence and in the replicative senescence of normal human cells [55]. Ras (mutant HRAS and NRAS)-induced senescence involved ROS and EGFR-argonaute 2 (AGO2) signaling in various human cancer cells, including HeLa, melanoma cells, and bladder cancer cells [56]. AGO2 bound to Ras and the loss of AGO2 inhibited cancer cell proliferation, senescence, and Ras signaling [56]. Thus, targeting AGO2 could eliminate senescent cancer cells. These reports suggest that inhibiting senescence induced during oncogenic transformation is necessary for overcoming tumorigenesis.

The immediate early response gene 2 (IER2) has been associated with the progression of several types of cancer. The sustained expression of IER2 induced senescence in a subset of melanoma cells in a p53/mitogen-activated protein kinase (MAPK)/AKT-dependent manner [57]. The senescent cells produced extracellular factors, such as osteopontin (OPN). OPN secreted by IER2-expressing senescent cells stimulated the migration and invasion of non-senescent melanoma cells [57]. The downregulation of OPN enhanced sensitivity to 5-fluorouracil (5-FU) in HepG2 cells [58]. This suggests that OIS leads to the secretion of extracellular factors that can promote cancer cell proliferation and anticancer drug resistance. Figure 3 shows the effects of oncogenes on senescence and the mechanisms of OIS.

## 4. Roles of SASP in Cancer Cell Proliferation and Therapeutic Resistance

Senescence inhibits the replication of damaged and precancerous cells in the short-term but is implicated in various diseases owing to SASP secretions. Senescent cells can induce other non-senescent cells to undergo senescence (secondary senescence) via SASP. Secondary senescence occurs via paracrine mechanisms and cell-to-cell contacts (juxtacrine) (Figure 1). The SASP includes pro-inflammatory cytokines, chemokines, MMPs, bioactive lipids, and growth factors (Figure 4) [59,60,61,62], as well as vascular endothelial growth factor (VEGF), hepatocyte growth factor (HGF), interleukins (IL-1a, IL-6, IL-8, IL-10, IL-13, IL-15), and CC/CXC chemokines (CXCL1, CXCL2, CXCL5, CXCL11, CXCL12, CCL2, CCL20) (Figure 4).

The SASP also includes extracellular vesicles (sEVs), which mediate cell-to-cell communication through various types of cargos, including proteins, lipids, and miRNAs. sEVs activate macrophages by increasing cellular levels of pro-inflammatory cytokines, such as IL-1β and IL-6, via NF-κB signaling [63]. sEVs released by senescent vascular smooth muscle cells (VSMCs) induced the secretion of IL-17, interferon γ (INFγ), and IL-10 by T cells, and tumor necrosis factor α (TNFα) by monocytes. sEVs released by senescent VSMCs promoted M1 macrophage polarization favoring the pro-inflammatory phenotype [64]. Thus, SASP could modulate immune responses. 

Ras and tumor protein 53 (TP53) mutations, and cellular senescence are frequently detected in zebrafish imaging [63]. Additional TP53 mutation changes OIS from a tumor suppressor to a driver [65]. Newly transformed cells with an RASG12V mutation became senescent and were eliminated from the epithelia. This was prevented by adding a TP53 gain-of-function mutation (TP53R175H). The surviving RASG12V-TP53R175H double-mutant cells secreted SASP-related inflammatory molecules that converted neighboring normal cells into SASP factor-secreting senescent cells, generating a heterogeneous tumor-like cell mass [65]. Thus, OIS involves the secretion of the SASP, which further spreads senescence through cellular interactions. 

Senescence and the associated SASP can affect cells within the tumor microenvironment (TME). The SASP was shown to promote tumorigenesis [66], EMT [67,68,69], cancer stemness [70,71], anticancer drug resistance [72,73,74], angiogenesis [75,76,77,78], immune suppression to promote tumorigenesis [79,80], and tumor suppression by the immune clearance of senescent cells [81]. Figure 4 shows the roles of the SASP, including tumor suppression and tumor promotion. 

The SASP increased the expression of lipocalin-2 (LCN2) to enhance breast cancer cell proliferation and migration [66]. High levels of LCN2 could predict the poor prognosis of patients with breast cancer [66]. The inactivation of LCN2 enhanced the response to chemotherapeutic drugs in mouse models of breast cancer [64]. This implies the roles of the SASP in cancer cell proliferation and anticancer drug resistance. Senescent lung fibroblasts (SLF) can promote the production of IL-6 and IL-8, which are effectively inhibited by overexpressing Klotho (Figure 4) [68]. Conditioned medium (CM) of SLFs enhanced the migration of lung cancer cells and increased the expression of the phospho-signal transducer and activator of T cell 3 (p-STAT3) but decreased the expression of P53 and E-cadherin [68]. Klotho inhibited the extracellular release of IL-6 and IL-8, which influenced STAT3 activation, P53 expression, and EMT in lung cancer cells, finally inhibiting lung cancer cell growth and migration [68]. CM from senescent MPM (malignant pleural mesothelioma) cells trigger, via STAT3 activation, the emergence of EMT-like, clonogenic, and chemo-resistant cell subpopulations, expressing high levels of ALDH (aldehyde dehydrogenase) activity (ALDH(bright) cells) [73]. Thus, SASP can enhance EMT, tumorigenic potential, and the anticancer drug resistance of senescent cancer cells.

Senescent fibroblasts promote angiogenesis by secreting VEGF, a component of the SASP [76]. The increased expression of VEGF resulted from senescence induced by Ras expression or p16 overexpression [76]. This suggests that senescence can also promote tumorigenesis. CM of IL-1β (a component of SASP)-treated chondrocytes enhanced migration and the tube-forming potential of human umbilical vein endothelial cells (HUVEC) in a growth differentiation factor 15 (GDF15)-dependent manner [78]. CDK4/6 inhibitors induced senescence and increased the production of pro-angiogenic SASP factors such as VEGF, basic FGF (bFGF), PDGF, as well as multiple MMPs, resulting in vascular remodeling in preclinical models of pancreatic ductal adenocarcinoma (PDAC) [82]. This vascular remodeling enhanced the efficacy of anti-PD-1 by improving gemcitabine delivery to the tumors and promoting cytotoxic T-cell infiltration [73]. This implies that TIS could enhance the efficacy of current anticancer therapy by inducing the SASP-promoted angiogenesis. 

Cancer cells emerging from senescence display higher anticancer drug resistance than cancer cells without senescence [46]. Senescent cells display resistance to apoptosis owing to high level of BCL-2 [83], suggesting the role of senescence in therapeutic resistance. Lapatinib and fulvestrant (EGFR and HER2 inhibitors) treatment induced apoptosis in chemotherapy-induced senescent breast cancer cells [84]. Senescence induced by radiation, oncogenes, and chemicals increased the expression of BCL-2 in non-transformed retinal pigment endothelial cells and fibroblasts [83]. Thus, targeting BCL-2 or other anti-apoptotic proteins can eliminate senescent cancer cells (senolysis or senolytic therapy).

IL-8, a component of the SASP, was necessary for the increased expression of multi-drug resistance 1 (MDR1)/ABCB1 by chemotherapy in urothelial carcinoma endothelial cells [85]. Targeting ABCB1 and/or IL-8 in tumor endothelial cells may overcome chemotherapeutic resistance and improve clinical outcomes. Low levels of IL-8 are correlated with longer overall patient survival [86]. The IL-8/STAT3 axis is necessary for hepatocellular carcinoma metastasis mediated by Snail and Twist 1 [86]. Anticancer drug resistance is closely related to the enhanced metastatic potential of cancer cells [87]. Since IL-8 is a component of the SASP, it is probable that the SASP may confer anticancer drug resistance. 

Cancer stem cells (CSCs) play a critical role in resistance to chemotherapy. Doxorubicin increased the expression of stemness-related genes, such as epithelial cell adhesion molecule (EpCAM), cytoskeletal 19 (CK19), annexin A3 (ANXA3), and the multidrug-resistance-related gene ABCG2 in hepatocellular carcinoma cells (HCCs) [71]. Platinum induced senescence and cancer stemness in ovarian cancer cells by enhancing aldehyde dehydrogenase 1A1 (ALDH1A1) activity [88]. These reports suggest that doxorubicin and platinum induced the SASP to promote cancer stemness and therapeutic resistance.

CD44 conferred doxorubicin resistance by upregulating MDR1 expression in osteosarcoma cells [89]. CD44, a cell surface marker of CSCs [90], conferred 5-FU resistance by increasing intracellular glutathione and suppressing the drug-induced production of ROS in gastric cancer cells [91]. Thus, the acquisition of cancer-stem-cell-like properties by senescence could induce anticancer drug resistance. The IL-6/STAT3 axis was necessary for chemo resistance in glioblastoma stem cells [92]. IL-6/IL-8-JAK2 signaling activated by cancer-associated fibroblasts (CAFs) conferred resistance to a bromodomain and extra-terminal (BET) inhibitor in colorectal cancer cells [93]. The NF-κB-IL-6-STAT3 signaling pathway increased cancer stemness and the tumorigenic potential of glioma cells [94]. Since IL-6/IL-8 is a component of the SASP, it is probable that the SASP may confer chemotherapeutic resistance by enhancing cancer-stem-cell-like properties. Erlotinib increased the expression of p21, p53, and IL-6 in keratinocytes [95]. Thus, erlotinib resistance could be induced by senescence. Erlotinib resistance involved the activation of STAT3 by EGFR signaling in osteosarcoma cells [96]. Since STAT3 can enhance cancer-stem-cell-like properties, erlotinib resistance may be associated with the enhanced cancer-stem-cell-like properties. These reports suggest the role of the SASP in anticancer drug resistance in relation to cancer stemness.

A persistent inflammatory state evokes compensatory immunosuppression, which impairs the functions of effector immune cells, such as macrophages, T cells, and natural killer (NK) cells. DNA damage induced the production of amphiregulin (AREG), a component of the SASP, in stromal cells, which increased the expression of immune checkpoint molecule programmed death-ligand 1 (PD-L1) in cancer cells and induced immune suppression in the TME to promote tumor formation [72]. PD-L1 is overexpressed in cancer cells and confers resistance to cancer therapy. Thus, the SASP can induce immune suppression to promote tumor progression. 

The SASP of senescent cancer cells enhanced immunosenescence (immune suppression) in the TME of peritoneal carcinomatosis (PC) to promote tumor progression [97]. Doxorubicin-induced senescence promoted the recruitment of Treg cells and myeloid-derived suppressor cells to mediate apoptotic resistance in diffuse large B-cell lymphoma (DLBCL) via pro-inflammatory cytokines, such as IL-2, IL-6, IL-8, IL-10, IL-35, TGFβ, and VEGF [77]. Senescent cancer cells were shown to mediate pro-tumorigenic effect by inhibiting the infiltration of CD8+ T cells via CXCL12/CXCR4 signaling [98]. Thus, the SASP of senescent cancer cells could promote cancer recurrence by exerting immune suppression.

IL-8 secreted by hypoxic macrophages enhanced esophageal cancer cell proliferation and metastasis by increasing the expression of PD-L1 in esophageal cancer cells [99]. PD-L1 expression was positively correlated with high levels of the SASP in a mouse model [100]. This implies that IL-8, a SASP component, could induce immune evasion by inhibiting CTL activity against PD-L1 expressing cancer cells. These reports imply the role of PD-L1 in senescence. PD-L1 depletion induced senescence by increasing the expression of stimulator of interferon genes (STING) in human melanoma cells and lung cancer cells [101]. This suggests that the blockade of PD-L1 could enhance the sensitivity of cancer cells to cancer therapeutics by enhancing senescence. Thus, the SASP could induce immune suppression by increasing the expression of PD-L1, which leads to tumor promotion. 

Senescent cancer cells in tumors are metabolically active and can promote tumor relapse, metastasis, and resistance to therapy through the SASP [46,102,103,104,105,106,107]. The pro-inflammatory SASP promoted the secretion of pro-tumorigenic factors, such as IL-1α and IL-6, and a less inflammatory SASP promoted the clearance of senescent cells (senescence surveillance) by the immune system [108]. This implies the role of the SASP in tumor suppression. Senescent cells can undergo conversion to an immunogenic phenotype that enables them to be eliminated by the immune system. If senescent cancer cells are not cleared, they acquire the potential to recapture the proliferative state. The impaired surveillance of pre-malignant senescent cells resulted in the development of murine hepatocellular carcinoma (HCC) [109]. Senescent cells were reported to secrete various inflammatory cytokines, such as IL-1β, IL-6, IL-8, TGF-β, granulocyte-macrophage colony-stimulating factor (GM-CSF), and MCP-1, to induce senescence surveillance in p16-positive myeloid cells [110]. During Ras-induced senescence, Cyclooxygenase-2 (COX-2) promoted senescence surveillance and tumor suppression by regulating multiple components of the SASP, such as prostaglandin E2 (PGE2) and IL-6 [81]. Thus, the SASP suppressed tumor progression by inducing senescence surveillance. 

## 5. Role of Autophagy in Senescence

Radiation-induced senescence involves increased autophagy in lung cancer cells [18]. Senescent cells appear large, flattened, and irregularly shaped, which is attributed to increased mammalian target of rapamycin (mTOR) signaling [111,112]. Derivatives of indolo [2,3-a]pyrrolo [3,4-c]carbazole induce senescence by modulating AKT/mTOR/S6K signaling in breast cancer cells, lung cancer cells, and colon cancer cells [112]. These reports suggest a close relationship between autophagy and senescence. Autophagy is a self-digesting cellular process that allows cells to sequester cytoplasmic contents through the formation of double membrane vesicles (autophagosomes). As with senescence, autophagy is stimulated by oxidative stress, chemotherapy, oncogene activation, and various other conditions [113]. Cellular senescence and autophagy commonly induce oxidative stress, DNA damage, telomere shortening, and oncogenic stress. Thus, senescence and autophagy are interrelated. Complete autophagy can lead to cell death. Defective autophagy is linked to various human diseases such as cancer, neurodegeneration, microbial infection, and aging [114]. SIRT3 promotes autophagy to suppress doxorubicin-induced senescence by inactivating the PI3K/Akt/mTOR pathway in lung cancer cells [115]. However, it is still unclear whether autophagy has a positive or negative effect on senescence.

Receptor tyrosine kinase inhibitors (RTKi) induced protective autophagy for cell survival and anticancer drug resistance in neuroblastoma cells [116]. Autophagy promoted the acquisition of cancer-stem-cell-like properties [117]. The inhibition of autophagy by siATG5 or 3-MA enhanced the sensitivity of liver cancer stem cells to doxorubicin [117]. It is interesting to examine whether autophagy can induce senescence in liver cancer cells. Resistance to clinically used anticancer drugs developed in prostate cancer cells and breast cancer cells due to autophagy in [118]. Thus, the inhibition of autophagy could overcome resistance to these clinically used anticancer drugs. Cancer/testis antigen CAGE promotes anticancer drug resistance by promoting autophagy in gastric cancer cells [119]. Icotinib resistance resulted from the increased expression of autophagic flux (ATG3, ATG5, and ATG7) via STAT3-FOXM signaling in lung cancer cells [120]. Chloroquine (CQ), an inhibitor of autophagy, enhanced the sensitivity of lung cancer patients to icotinib [120]. CAGE was shown to bind to Beclin1 (a mediator of autophagy) and confer resistance to various anticancer drugs in non-small-cell lung cancer cells (NSCLCs) [121]. This suggest that autophagy promotes anticancer drug resistance. 

Erlotinib induced autophagy to promote erlotinib resistance via 6-phosphofructo-2-kinase/fructose-2,6-biphosphatase 3 (PFKFB3) in NSCLCs [122]. Wingless-related integration site 5A (Wnt5A)-induced autophagy mediated doxorubicin resistance in osteosarcoma cells [123]. The inhibition of autophagy sensitized glioblastoma cells to Src tyrosine kinase inhibitors such as Si306 and pro-Si306 [124]. These reports suggest that targeting autophagy may enhance the sensitivity of cancer cells to anticancer drugs. These reports also imply that anticancer drugs induce therapy resistance by promoting autophagy in association with senescence.

The inhibition of STAT3 restored sensitivity to icotinib by reducing autophagy [120]. Since STAT3 regulates the expression of IL-6 and IL-8, components of the SASP, it is probable that the SASP may regulate resistance to icotinib in association with senescence. Autophagy induction during mitotic slippage involved the autophagy activator AMP-dependent kinase (AMPK) and the endoplasmic reticulum stress response protein kinase R-like ER kinase (PERK) [125]. The pharmacologic inhibition of autophagy or downregulation of ATG5 led to a bypass of G1 arrest senescence, reduced SASP-associated paracrine tumorigenic effects, and increased DNA damage after S-phase entry with a concomitant increase in apoptosis in various cancer cells, including osteosarcoma cells, colon cancer cells, pancreatic cancer cells, and breast cancer cells [125]. These reports suggest that autophagy can promote senescence.

Oncogene HRAS-induced senescence involved the upregulation of genes involved in autophagy, such as unc51-like autophagy activating kinase 3 (ULK3), in human diploid fibroblast IMR90 cells [126]. HRAS-induced senescence was negatively regulated by the PI3K-mTOR signaling pathway [124]. Hyperglycemia (HG) (25 mM) increased the expression of Beclin-1, ATG 5, 7, and 12, the generation of microtubule-associated proteins 1A/1B light chain 3B (LC3-II), and autophagosome formation that was correlated with the development of cellular senescence [127]. N-acetyl-L-cystein (NAC) inhibited the effect of HG on senescence by decreasing ROS formation (Figure 5) [127]. X-ray radiation induced senescence and increased the expression of STAT3, Beclin1, and the LC3-II/LC3-I ratio in lung adenocarcinoma cells [18]. These reports suggest a crosstalk between autophagy and senescence. Figure 5 shows that complete autophagy may induce cell death, while impaired autophagy induces senescence. Figure 5 also shows the mechanisms of autophagy-induced senescence.

## 6. MicroRNAs in Senescence

MiRNAs, noncoding RNAs, are families of 21–25 nucleotides and small RNAs. These RNAs negatively control gene expression by binding to the 3′ untranslated region (UTR) of mRNA. Since miRNA targets multiple genes, miRNAs are involved in various diseases. Senescence increases the secretion of EVs, components of the SASP, and EVs can transport proteins and miRNAs. Since the SASP mediates cellular interactions, miRNAs may play critical roles in the spread of senescence. miR-34a targets Wnt and Notch pathways, as well as BCL2, and was found in EVs secreted from subjects with senescence [128]. Thus, targeting miR-34a may improve senescence. miR-485-5p targeted CDK16 to decrease the expression of C-MYC and PD-L1, which, in turn, induced the senescence-associated phenotypes in lung cancer cell lines [129]. Therefore, CDK16 can be a target for the induction of senescence in cancer cells. Thus, an miR-485-5p mimic could be employed for inducing senescence in cancer cells.

Doxorubicin increased the number of senescence-associated secretory phenotypes but decreased the expression of miR-199a-3p in cardiomyocytes [130]. The overexpression of miR-199a-3p inhibited SASP production and the spread of senescence by decreasing the expression of GATA-4 in cardiomyocytes [130]. This suggests that an miR-199a-3p mimic could be developed as an anticancer drug owing to its role in regulating SASP production.

Levels of miR-200 family members are decreased in the late stage of tumorigenesis [67]. miR-200 full knock out (FKO)-associated senescence in cancer epithelial cells recruited stromal cells in the TME [65]. The genetic deletion of all miR-200s in human gastric cancer cell lines induced potent morphological alterations and G1/S cell cycle arrest, increased SA-β-gal activity, and activated TGF-β/TNF-α signaling and aberrant metabolism, collectively resembling the senescent phenotype [67]. MiR-200b is negatively regulated by CAGE in anticancer drug-resistant melanoma cells [131]. Thus, CAGE can confer anticancer drug resistance by inducing senescence in melanoma cells by decreasing miR-200b expression. Thus, inhibitors of the miR-200 family can enhance the effects of anticancer drugs by inducing senescence.

The overexpression of miR-335 and miR-34a induced the premature senescence of young mesangial cells via the direct suppression of superoxide dismutase 2 (SOD2) and thioredoxin reductase 2 (Txnrd2), with a concomitant increase in ROS in mesangial cells [132]. ROS induced DNA damage in osteosarcoma cells, which, in turn, increased miR-335 expression in a p53-dependent manner [133]. miR-335 and p53 cooperated in a positive feedback loop to drive cell cycle arrest in osteosarcoma cells [133]. Thus, miR-335 activation plays an important role in the induction of p53-dependent senescence after DNA damage [133] and an miR-335 mimic could enhance the effects of anticancer drugs by inducing senescence in cancer cells.

The overexpression of miR-217 promoted vascular endothelial cell senescence by targeting the NAD-dependent deacetylase sirtuin 1 (SIRT1)/p53 signaling pathway [134]. MiR-21-5p/miR-217 carried by sEV from senescent cells spread pro-senescence signals by targeting DNA methyl transferase 1 (DNMT1) and NAD-dependent deacetylase sirtuin 1 (SIRT1) and inhibiting endothelial cell proliferation [135]. Thus, it is probable that an miR-217 mimic could prevent endothelial cells from promoting cancer cell proliferation. 

MiR-335, upregulated in senescent normal fibroblasts and CAFs, modulated the secretion of SASP factors and enhanced the invasion potential of squamous cell carcinoma in co-cultures by suppressing the expression of phosphatase and tensin homolog (PTEN) [136]. Additionally, elevated levels of COX-2 and PGE2 were observed in senescent fibroblasts, and the inhibition of COX-2 by celecoxib reduced the expression of miR-335, restored PTEN expression, and decreased the pro-tumorigenic effects of the SASP [136]. miR-335 mediated the pro-tumorigenic effect of the SASP by decreasing the expression of PTEN and SASP [136]. It is probable that an miR-335 inhibitor could be developed as an anticancer drug that could enhance current anticancer therapy. Table 1 shows the roles of miRNAs in senescence and the mechanisms.

## 7. Role of HDACs in Senescence

Histone modifications regulate gene expression. Tet methylcytosine dioxygenase 2 (TET2) inhibited PD-L1 gene expression by recruiting histone deacetyalses (HDACs) (HDAC1 and -2) in MDA-MB-231 cells [137]. This implies that HDACs can regulate the immune evasion of cancer cells. Since the SASP regulates immune evasion [79], it is probable that HDACs may regulate senescence. Histone methylation (H3K27me3) in the promoter region of cell-cycle-dependent kinase N2a (CDKN2a) by enhancer of zeste homolog 2 (Ezh2), a histone methyl transferase, inhibited replicative senescence in atrial fibroblasts [138]. These reports suggest the roles of histone modifications in senescence. 

4-Phenylbutyric acid (4-PBA), an HDAC inhibitor (HDACi), induced EMT in gastric cancer cell lines by upregulating IL-8 [139]. IL-8, a component of the SASP, is known to mediate immune suppression and the recruitment of MDSCs (Figure 4). This further suggests the role of HDACs in senescence. SAHA (vorinostat), an HDACi, or the specific downregulation of HDAC2 or HDAC7 by siRNA, induced senescence in dermal fibroblasts by increasing the expression of IL-6, IL-8, MMP-1/-3, and IL-1β [33]. The ectopic overexpression of HDAC7 by lentiviral transduction in pre-senescent dermal fibroblasts extended their proliferative lifespan [33]. These reports suggest that HDACs could inhibit senescence. 

HDAC1/2-deficient epidermis displayed elevated acetylated p53 and the increased expression of the senescence gene p16 in pre-basal cell carcinoma [140]. This suggests the roles of HDAC1/2 as negative regulators of senescence. CAGE was shown to bind to HDAC2 and directly decreases the expression of p53 and confers resistance to anticancer drugs in melanoma cells [141]. This suggests that HDAC2 may confer resistance to anticancer drugs by suppressing p53-mediated senescence. HDAC2-specific inhibitors may enhance the effects of anticancer drugs by inducing senescence. The overexpression of HDAC2 inhibited cigarette smoke (CSE)-induced senescence by decreasing muscle ring finger protein 1 (MURF1), ATROGIN1, P53, and P21 expression in C2C12 cells [142]. These reports suggest the role of HDAC2 in senescence, cancer cell proliferation, and anticancer drug resistance.

HDAC6 differs from the other HDAC family members not only in terms of its cytoplasmic localization but also in terms of its substrate repertoire and hence cellular functions. HDAC6 mostly deacetylates cytoplasmic proteins [143]. An HDAC6 knockout mice study showed the role of HDAC6 in cell survival under stress conditions [144]. The downregulation of HDAC6 inhibited the proliferation and migration of glioblastoma cell lines and the sonic hedgehog (shh) pathway, and decreased autophagy [145]. ACY-1215 (ricolinostat, a specific HDAC6 inhibitor) suppresses glioblastoma growth by inhibiting TGF-β/SMAD2 signaling to decrease the expression of p21 [146]. This suggests the role of HDAC6 in promoting senescence. Given the fact that HDAC6 regulated autophagy [145], HDAC6 could induce senescence by promoting autophagy.

HDAC6 inhibitors decreased the expression of PD-L1 in urothelial cancer cell lines [147]. Since SASP regulates the expression of PD-L1 [72,99,100], HDAC6 inhibition can suppress the immune evasion of cancer cells by decreasing autophagy and senescence. High levels of HDAC6 are observed in colon cancer patients with KRAS activation mutation [148]. Oncogenic KRAS contributes to SAHA resistance by upregulating HDAC6 and C-MYC expression [148]. Oncogenes can induce senescence (Figure 2). It is probable that K-RAS enhances SAHA resistance by promoting autophagy and senescence. The acetylation of lysine 104 by HDAC6 inhibition attenuated KRAS-transforming activity by interfering with guanine nucleotide exchange factor (GEF)-induced nucleotide exchange in various cancer cells [149]. Thus, targeting HDAC6 may overcome the oncogenic effects of KRAS by regulating senescence. These reports implicate the role of HDAC6 in autophagy and senescence. 

Azaindolyl sulfonamide, a potent HDAC6 inhibitor, induced G2/M arrest and senescence in temozolomide-resistant glioblastoma cells [150]. Microtubule stability was increased during senescence by enhancing α-tubulin acetylation through the depletion of HDAC6 [151]. These reports imply that HDAC6 may act as a negative regulator of senescence. The natural product 2,4-di-tert-butylphenol (DTBP), an HDAC6 inhibitor, induced senescence (mitotic catastrophe) in human gastric adenocarcinoma AGS cells by increasing the expression of p21 and Rb [152]. Thus, DTBP targets HDAC6 to induce senescence. HDAC6 was increased by NF-κB and promoted hepatocellular carcinoma cell proliferation by inducing the degradation of p53 in HCCs [153] and promoted the invasion of breast cancer cells by decreasing E-cadherin while increasing STAT3 levels [154]. These reports suggest that HDAC6 promotes cancer cell proliferation by inhibiting senescence. It is probable that HDAC6 helps cancer cells escape senescence surveillance to display aggressive phenotypes. Thus, the inhibition of HDAC6 could enhance the effects of anticancer drugs by inducing senescence. 

Ras promotes cisplatin resistance through HDAC4, which activates transcription factor CREB3 regulatory factor (CREBZF) to increase ATG3 expression [110]. This suggests the positive regulatory role of HDAC4 in autophagy and anticancer drug resistance. HDAC4 was highly expressed in CRC tissues, while p53 was poorly expressed in CRC tissues [155]. The downregulation of HDAC4 increased the expression of p53 in CRC [155]. This implies the negative regulatory role of HDAC4 in senescence. High HDAC4 levels predicted the poor overall survival (OS) of ovarian cancer patients [156]. The downregulation of HDAC4 was seen in aged or UV-irradiated skin [157]. The overexpression of HDAC4 overcame senescence by increasing the expression of DNA-damage-inducible transcript 4 (DDI4) [157]. HDAC4 was degraded during senescence via ubiquitination, and the deletion of HDAC4 in transformed mouse embryonic fibroblasts (MEF) was responsible for the reappearance of senescence by enhancing H3K27 acetylation levels [158]. These reports suggest that HDAC4 may act as a negative regulator of senescence. 

An HDAC4 triple mutant mediated OIS in a p53-dependent manner in human primary fibroblasts [159]. HDAC4 induced growth arrest and DNA damage was seen from an unusual pattern of γH2AX-positive foci [159]. Thus, HDAC4 can act as a positive regulator of senescence. Since HDAC4 regulates senescence and autophagy, it is reasonable to employ HDAC4 as a target for developing anticancer drugs. Taken together, HDACs regulate senescence in a context-dependent manner in association with autophagy. Table 2 shows the roles of HDACs in senescence.

## 8. Conclusions and Perspectives

Since senescent cancer cells can evade immune surveillance (senescence surveillance) and acquire aggressive phenotypes, it is necessary to eliminate those senescent cancer cells. For this, a one-two punch strategy has been tried. The first step involves the induction of senescence with chemotherapy or other treatment modalities. The therapeutic induction of senescence is a potential means to treat cancer, primarily acting through the induction of a persistent growth-arrested state in tumors. Defensin-like peptide CopA3 increased SA-β-gal activity and ROS levels and DNA damage and G1 cell cycle arrest in colon cancer cells [160]. Thus, CopA3 is useful for the induction of senescence in cancer cells. CopA3 can be used with current chemotherapeutic drugs to eliminate senescent cancer cells. The peptides that can increase the expression of p53 or other inducers of senescence may enhance the sensitivity of cancer cells to current anticancer drugs. 

The inflammatory cytokines released by senescent cells can upregulate PD-L1 in non-senescent cells via the JAK-STAT pathway [161]. This suggests that the SASP can confer an immune checkpoint response in non-senescent cells. Chemotherapeutic drugs that can increase the expression of PD-L1 induce senescence and can sensitize cancer cells to anticancer drugs. Doxorubicin-induced senescence promoted the recruitment of PD-L1-expressing T cells [162]. Thus, the induction of senescence by chemotherapy may enhance the immunotherapeutic efficacy of anti-PD-L1 antibodies. Therefore, senescence can serve as a target for developing anticancer drugs employing anti-PD-L1 antibodies. Chemotherapy induces cellular senescence in lung cancer cells and the evasion of senescence surveillance occurs via the activation of Cdc2/Cdk1 [163]. Thus, senescence-inducing chemotherapeutic drugs can be used with inhibitors of Cdc2/Cdk1. 

The second step consists of follow-up treatment with senolytic drugs. Senolytic drugs selectively eliminate senescent cells to attenuate senescence-mediated pathologies. The report on the one-two punch approach to selectively eliminate chemotherapy-induced senescent lymphoma cells in mice was achieved by using a metabolic senolytic to block glucose utilization or autophagy [164]. The elimination of senescent lymphoma occurs through caspase-12- and caspase-3-mediated endoplasmic-reticulum-related apoptosis [164]. Nano-drugs, such as lapatinib- poly(amidoamine) and fulvestrant- poly(amidoamine), promoted the elimination of doxorubicin-induced senescent breast cancer cells by apoptosis-mediated senolysis [84]. Thus, the induction of senescence can sensitize cancer cells to senolytic drugs.

The main task of senolytic drugs is the elimination of senescence cells by targeting anti-apoptotic pathways. Panobinostat, a Food and Drug Administration (FDA)-approved histone deacetylase inhibitor, eliminated TIS cancer cells (NSCLCs and head and neck squamous cell carcinoma) by decreasing the expression of BCL-XL [165]. Panobinostat has shown an antitumor effect by inhibiting Wnt/β-catenin signaling in breast cancer [166]. Since TIS involves the increased expression of BCL-2, targeting BCL-2 may exert anticancer effects. BCL2-targeting drugs, including ABT-737 and ABT-263 (navitoclax), natural substances, such as artesunate, fisetin, and curcumin, showed senolytic effects in glioblastoma cells and non-transformed cells [83,167]. ABT-263 eliminated senescent cells and enhanced the efficacy of olaparib, a PARP inhibitor, in both breast and ovarian cancer cell lines [168]. Olaparib increased the expression of Bcl-XL, inflammatory cytokines, check point kinase 2 (Chk2), and p21 [168]. Thus, PARPi and senolytic drugs can induce efficient clinical outcomes. Oridonin, an active ingredient of Chinese herbal medicine, targeted glutathione *S*-transferases to activate ROS-p38 signaling and eliminate senescent cell via apoptosis [169]. Doxorubicin induced senescence in lung cancer cells by increasing cytoplasmic p21 and the senolytic effect of ABT-263 was enhanced in p21 knockout lung cancer cells [170]. 

Senolytic drugs, such as dasatinib, quercetin, and fisetin, eliminated doxorubicin-induced senescent cells by inducing apoptosis [30]. Doxorubicin increased the expression of the SASP to induce senescence in endothelial cells [30]. Dasatinib and quercetin attenuated cisplatin-induced ovarian injury by targeting senescent cells and reducing DNA damage in primary granulosa cells [171]. Fisetin induced G2/M arrest and apoptosis in NSCLCs by inhibiting EGFR/ERK1/2/STAT3 signaling [172]. Thus, TIS can sensitize cancer cells to senolytic drugs. The identification of the targets of these senolytic drugs may help us to better understand the mechanisms of senescence. Although senolytic drugs have shown some promising anticancer effects, current drugs have some limitations. Since senolytic drugs target aging and aging-related diseases, they might cause undesirable side effects. Senolytic drugs cause tissue atrophy through the massive removal of senescent cells. The complete removal of senescent cells wipes out the beneficial effects of senescent cells, including tissue regeneration, metabolic reprograming, and wound healing. It will be necessary to target another aspect of senescence and increase the specificity of senolytic drugs.

OIS causes G1 cell cycle arrest and limits the generation of cancer stem cells [173]. Since oncogenes can induce senescence, targeting oncogenes can be employed for overcoming OIS and anticancer drug resistance. Wogonin, a natural flavonoid compound, induced cellular senescence in T-cell malignancies and activated the p53-mediated DNA damage response [26]. The upregulated expression of BCL-2 was seen in senescent malignant T cells via the anti-apoptotic properties of senescent cells [28]. ABT-263 induced apoptotic cell death in wogonin-induced senescent cells [28]. Thus, the induction of senescence by wogonin increased the sensitivity of malignant T cells with a low expression of BCL-2 to ABT-263. Although oncogenes can induce senescence, they can also promote pathways that can inhibit senescence. Thus, it is necessary to identify oncogene-activated pathways that can promote or inhibit senescence. 

Selectively inhibiting the SASP in normal cells comprising the TME by senostatics may enhance the therapeutic efficacy of current therapies by eliminating pro-tumorigenic factors. Silybum marianum flower extract (SMFE) suppressed the expression of SASP factors such as IL-6 and MMP-1 in senescent human diploid fibroblasts (HDFs) [174]. The CM of HDFs treated with SMFE decreased the expression of p16 and SASP in keratinocytes [174]. Rapamycin, an inducer of autophagy, inhibits senescence and SASP by decreasing SA-β-Gal activity and p16 expression in human coronary artery endothelial cells [175]. Senostatics target the SASP regulatory network (NF-κB, p38 MAPK, mTOR, and others) or a specific component of the SASP. Components of the SASP regulatory network have other functions. Since the SASP shares many overlapping factors, it would be necessary to identify the SASP in a precise way. For this, a large-scale RNA interference screening may identify targets for senostatic drugs. An analysis of senescent human fibroblast cells revealed that individual senescent cells showed heterogeneity in their gene expression signatures [176]. This heterogeneity led to a weak correlation among genes encoding the SASP [176]. The composition of the SASP shows variation depending on the stimuli, the cell type, and the stage of senescence [176]. Therefore, it will be necessary to categorize SASP factors according to their functions in a context-dependent manner. 

Since HDACs play a role in senescence induction, it is necessary to examine whether HDAC inhibitors or activators can synergistically enhance the anticancer effects of senolytic drugs. The structural identification of these HDACs may help to identify novel drugs that enhance the anticancer effects of senolytic drugs. miRNAs have also been shown to regulate senescence. Therefore, miR mimics or miR inhibitors can be used in combination with senolytic drugs for the elimination of senescence-induced cancer cells.

The elimination of SASP-producing senescent cells or modification of the SASP could be the main areas of cancer therapy. However, it is necessary to further identify the factors responsible for the induction of senescence. This will make it possible to screen senescence-targeting anticancer drugs. Since the SASP is dependent on cell type, tissue, and the type of stress, it is necessary to define the SASP. Senescence-targeting drugs can synergistically enhance the anticancer effects of current cancer therapeutics. Overall, research on senotherapies is an extremely important area of cancer research that can reduce the adverse effects of senescence and improve current cancer therapies.

## Figures and Tables

**Figure 1 ijms-24-03436-f001:**
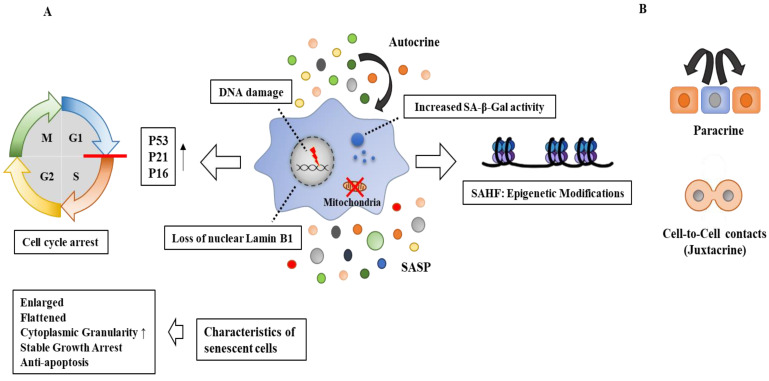
Characteristics of senescence. (**A**) SAHF is senescence-associated heterochromatin foci. Senescence is associated with SAHF in which expression of growth-related genes, such as targets of E2F, is suppressed. SASP refers to the senescence-associated secretory phenotype. Increased SA-β-gal activity leads to increased lysosomal mass. ↑ indicates an increase in expression/activity. The hollow arrows indicate the direction of the reaction. (**B**) Senescence is transmitted in paracrine and juxtacrine fashions. Senescent cells spread senescence via the SASP.

**Figure 2 ijms-24-03436-f002:**
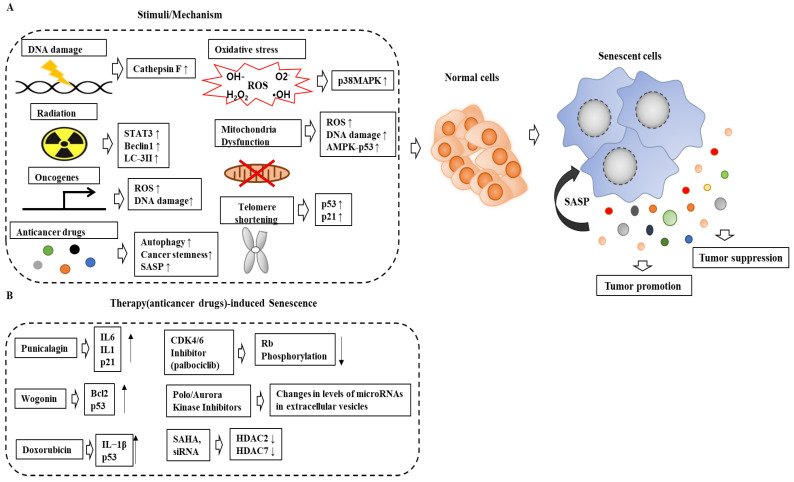
Factors inducing senescence and the mechanisms of senescence are shown. (**A**) Various factors induce senescence. Senescence initially leads to tumor suppression. Senescent cells can also promote tumor progression by recruiting immune suppressor cells, such as Treg cells and myeloid-derived suppressor cells (MDSCs). SAHA, an inhibitor of HDACs, is suberohydroxamic acid. ↑ indicates an increase in expression/activity. ↓ indicates a decrease in expression/activity. The hollow arrows indicate the direction of the reaction. (**B**) The mechanisms of therapy-induced senescence are also shown.

**Figure 3 ijms-24-03436-f003:**
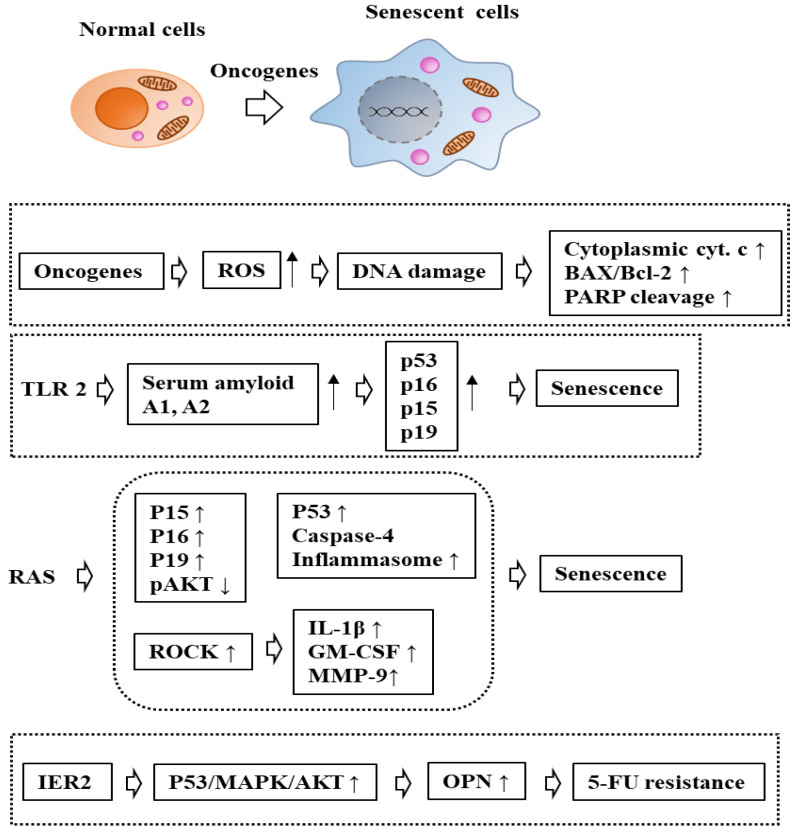
Mechanism of OIS. ↑ indicates an increase in expression/activity. ↓ indicates a decrease in expression/activity. The hollow arrows denote the direction of the reaction. OPN is osteopontin. TLR2 is toll-like receptor 2, and IER2 is immediate early response.

**Figure 4 ijms-24-03436-f004:**
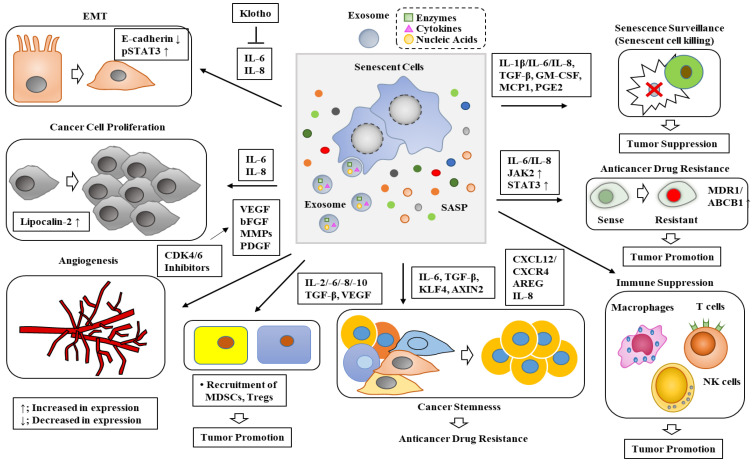
Roles of the SASP in tumor promotion and tumor suppression. The SASP of senescent cells regulates various life processes in a paracrine fashion. The SASP eliminates senescent cancer cells (senescence surveillance). The SASP exerts immune suppression to promote tumor progression. The SASP contains various cytokines/chemokines, MMPs, and soluble extracellular vesicles (EVs) to promote these activities. The hollow arrows and black arrows indicate the direction of reaction. → indicates the direction of the reaction. ↑ indicates an increase in expression/activity. ↓ indicates a decrease in expression/activity. The T bar indicates a negative regulation.

**Figure 5 ijms-24-03436-f005:**
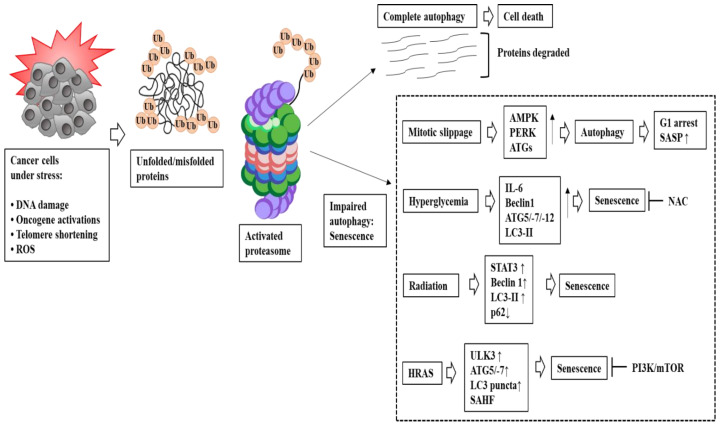
Autophagy leads to the induction of senescence. Autophagy is induced in cancer cells under various stresses. Complete autophagy usually results in cell death. Incomplete autophagy leads to senescence. ↑ indicates an increase in expression/activity. ↓ indicates a decrease in expression/activity. Hollow arrows and black arrows denote the direction of the reaction. The T bar indicates negative regulation. ULK3 is unc51-like autophagy activating kinase 3 and Ub is ubiquitin.

**Table 1 ijms-24-03436-t001:** Roles of miRNAs in senescence and the mechanisms.

miRNAs	Effect on Senescence	Mechanism	Reference
miR-34a	Promotes senescence	Targets Wnt/Notch/BCL-2	[128]
miR-485-5p	Promotes senescence (tumor-suppressive functions)	Targets CDK16 C-MYC ↑ PD-L1 ↑	[129]
miR-199a-3p	Inhibits senescence	SASP ↓	[130]
miR-200 family	Inhibits senescence	EMT ↓	[67]
miR-335/-34a	Promotes premature senescence	SOD ↓ Txnrd2 ↓ ROS ↑	[132]
miR-335	Promotes DNA-damage-induced senescence	Rb1 ↓ Forms a positive feedback loop with p53	[133]
miR-217	Promotes vascular endothelial cell senescence	Targets SIRT1/p53 signaling	[134]
miR-21-5p/miR-217	Promotes endothelial cell senescence	Targets DNMT1 and SIRT1	[135]
miR-335	Promotes senescence, cancer cell motility	PTEN ↓ SASP ↑	[136]

↓ indicates a decrease in expression/activity. ↑ indicates an increase in expression/activity. Rb1 is retinoblastoma 1.

**Table 2 ijms-24-03436-t002:** Roles of HDACs in senescence and the mechanisms.

HDAC	Effect on Senescence	Mechanism	Cells	Reference
HDAC2/ HDAC7	Inhibits senescence	RNAi of HDAC7 increases levels of IL-6, IL-8, and MMP-3	Dermal fibroblasts	[33]
HDAC2	Inhibits senescence	MURF1 ↓ ATROGIN1 ↓ P53 ↓, P21 ↓	C2C12 cells	[142]
HDAC6	Inhibits senescence	Microtubule stability ↑ ROCK ↓	Kidney epithelial cells	[150]
HDAC4	Inhibits senescence	DDI4 ↑	Dermal fibroblasts	[157]
HDAC4	Promotes senescence	TP53 ↑	Primary fibroblasts	[159]

↓ denotes a decrease in expression/activity; ↑ denotes an increased expression/activity. MMP-3, matrix metalloproteinase-3; RNAi, RNA interference; ROCK, Rho-associated protein kinase; TP53, tumor protein 539.

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
