# Peer review of "The Potential of Senescence as a Target for Developing Anticancer Therapy"

_ijms, 2023, doi:10.3390/ijms24043436_

Round 1
Reviewer 1 Report
In the manuscript, titled "The Potential of Senescence as a Target for Developing Anticancer Therapy", the authors intended to identify different signaling pathways to induce cellular senescence and suggested it as an anticancer therapeutic option. The research topic is very interesting; however, the manuscript has several key issues which needs to be clear before final publication.
1. There are several pros and cons of senescence as a cancer therapeutics. It will be interesting to discuss those points in the study. Oncogenes which are often regulated to induced senescence are still dangerous for multiple reasons. Certain SASP can also lead tumors progression.
2. Why immune cells are incapably of clearing senescence cells from the tumor microenvironment? It is very important to find ways to remove these harmful senescence cells.
3. Now a days, peptide therapy is evolving to target and for the clearance of these senescence cells. For example: FOXO4-DRI peptide was designed in such a way that it can efficiently inhibits the interaction between FOXO4-TP53 complex, thus triggers TP53-mediated apoptosis in senescence to kill these cells more easily.
4. Most of the time authors have directly coated the findings of a research paper, rather discussing it. For example: Line no. 172: ‘IL-8 is necessary for increased expression of multi drug resistance (MDR1)/ABCB1 by chemotherapy in urothelial carcinoma endothelial cells”. It is a finding of a study on a particular cell line. Authors need to discuss the findings and its relevance to the topic.
5. Figures needs modification in terms of detailing and new information. All the figures are very general.
6. The following articles are suitable for citation:
a. Baar et al., 2017. Cell. PMID: 28340339; PMID: 28340339.
b. Dey et al., 2021. Mech Ageing Dev. PMID: 33957217.
c. Braig et al., 2006. Cancer Res. PMID: 16540631.
Author Response
Dear Sir
Thanks for excellent suggestions. I made changes to accommodate your suggestions. I also sought help from professionals to fix English problems. I send English certificate. I hope that changes I made are fine.
Sincerely yours
Jeoung Dooil
Professor of Biochemistry
Kangwon National University
Chuncheon 24341, Korea
In the manuscript, titled "The Potential of Senescence as a Target for Developing Anticancer Therapy", the authors intended to identify different signaling pathways to induce cellular senescence and suggested it as an anticancer therapeutic option. The research topic is very interesting; however, the manuscript has several key issues which needs to be clear before final publication
Q1. There are several pros and cons of senescence as a cancer therapeutics. It will be interesting to discuss those points in the study. Oncogenes which are often regulated to induced senescence are still dangerous for multiple reasons. Certain SASP can also lead tumors progression.
Ans. I agree. Thanks for excellent suggestions.
- 〃It is probable that targeting oncogenes can eliminate senescent cells, which prevents transformation of senescent cells into cancer cells〃 (lines 112-114). Since oncogene can induce senescence, targeting oncogene can prevent emergence of malignant cancer cells from senescent cells.
- It is also noteworthy that oncogenes that induce senescence can also promote tumor initiation by inducing immune evasion [40]. Please take a look at lines 125-126. This is the reason why oncogenes can be dangerous.
- Transformed cells after senescence become more aggressive and display high invasion potential and anticancer drug resistance [38]. A combination of oncogenes (CTNNB1, TERT, and MYC) induced senescence in human fibroblasts and primary hepatocytes [39]. MYC is critical for the immune evasion of triple-negative breast cancer cells (TNBCs) [40]. TNBCs with high MYC are resistant to immune checkpoint inhibitor (ICI) therapy [40]. IL-6, a component of SASP, mediates the effect of MYC on somatic cell reprogramming (transformation) [41]. MYC is necessary for the SASP-induced somatic cell programming, such as generation of induced pluripotent stem cells [41]. It is probable that targeting oncogenes can eliminate senescent cells, which prevents transformation of senescent cells into cancer cells. Please take a look at lines 106-114.
- Senescent cells, via the SASP, induce immune suppression by inhibiting CD8+ T cells, macrophages, and NK cells (Figure 4). This immune suppression causes tumor promotion. Therefore, targeting SASP may prevent emergence of aggressive cancer cells from senescent cells.
- In some cases, the SASP induces senescence surveillance to promote tumor suppression (Figure 4, ref.102)â‘¥ . 〃Senescent cells were reported to secrete various inflammatory cytokines, such as IL-1β, IL-6, IL-8, TGF-β, granulocyte-macrophages colony-stimulating factor (GM-CSF), and MCP-1, to induce senescence surveillance in p16-positive myeloid cells [102]〃. Please take a look at lines 308-311.
- It is known that the SASP promotes tumor progression. Senostatic drugs can target the SASP to suppress tumor progression. I mention about senostatic drugs (lines 685-692).
- Why immune cells are incapably of clearing senescence cells from the tumor microenvironment? It is very important to find ways to remove these harmful senescence cells.
Ans. It is probable that senescent cells, via SASP, exerts immune suppression, which prevents immune cells from killing senescent cells. Cancer cells emerging from senescence display higher anticancer drug resistance than cancer cells without senescence [38]. It is therefore necessary to eliminate senescent cells. Senescence surveillance removes senescent cells and senescent cancer cells. Senescent cells secrete the SASP to induce senescence surveillance for senescent cell killing (Figure 4). IL-1β, IL-6, IL-8, TGF-β, GM-CSF, MCP1 and PGE2 promote senescence surveillance (Figure 4). Senescence surveillance is necessary for preventing emergence of malignant cancer cells from senescent cells. I mention about senescence surveillance. Please take a look at lines 307-311.
- Now a days, peptide therapy is evolving to target and for the clearance of these senescence cells. For example: FOXO4-DRI peptide was designed in such a way that it can efficiently inhibits the interaction between FOXO4-TP53 complex, thus triggers TP53-mediated apoptosis in senescence to kill these cells more easily.
Ans. I agree. I mention the effect of peptide that inhibits interaction between FOXO4 and TP53 on apoptosis in senescence. Please take a look at lines 694-698. I also add reference on FOXO4-DRI peptide (ref. 173).
Q4. Most of the time authors have directly coated the findings of a research paper, rather discussing it. For example: Line no. 172: ‘IL-8 is necessary for increased expression of multi drug resistance (MDR1)/ABCB1 by chemotherapy in urothelial carcinoma endothelial cells”. It is a finding of a study on a particular cell line. Authors need to discuss the findings and its relevance to the topic.
Ans. Thanks, I agree. I made significant changes throughout the manuscript. In this revision,
I change 〃IL-8 is necessary for increased expression of multi drug resistance (MDR1)/ABCB1 by chemotherapy in urothelial carcinoma endothelial cells〃into:
[IL-8, a component of the SASP, was necessary for the increased expression of multi drug resistance 1 (MDR1)/ABCB1 by chemotherapy in urothelial carcinoma endothelial cells [77]. Targeting ABCB1 and/or IL-8 in tumor endothelial cells may overcome chemotherapeutic resistance and improve clinical outcomes.]. Please take a look at lines 236-239.
I try to discuss each finding in detail throughout the manuscript. Please take a look at new manuscript.
Q5. Figures needs modification in terms of detailing and new information. All the figures are very general.
Ans. I Agree. Thanks for excellent suggestion. In this revision, I try to make changes in each figure to give more information. Please take a look at new figures. I think that each new figure gives more information.
Q6. The following articles are suitable for citation:
- Baar et al., 2017. Cell. PMID: 28340339; PMID: 28340339.
- Dey et al., 2021. Mech Ageing Dev. PMID: 33957217.
- Braig et al., 2006. Cancer Res. PMID: 16540631.
Ans. I add the above articles in new references.
Baar et al; reference 173
Dey et al; reference 159
Braig et al; reference 37
Please take a look at new references.
Reviewer 2 Report
This review addresses an important topic, the potential of senescence as an anti-cancer therapy. The authors have made a significant effort describing, often very briefly, the conclusions of many publications. However, the result of this is that the narrative is very difficult to follow as not enough background is provided for the reader to understand the specific conclusions. Sometimes, there is just one concluding sentence and the reader wonders whether the experimental design was in cells or in vivo in mice or humans etc. The authors could provide further background and group their conclusions, e.g evidence in vitro, in vivo in mice and humans, or in any other way to compartmentalise the narrative so it is better structured and organised.
The focus on autophagy, HDACs and miRNAs is not explained, and the reader wonders why to chose these three subjects. I do not find these particularly interesting or relevant. However, I miss sections discussing the field of senotherapies; senolytics, senomorphics, biotherapeutics, PROTACs, pro-drugs, etc
Overall, I feel that the subject is very interesting but the current text needs extensive revision to improve the structure and background supporting the conclusions. I also find that the authors have cited evidence after evidence without these being assessed; in other words, the authors' vision or critical assessment of the evidence is currently missing. This is usually an important part of a review that helps synthesise the evidence and facilitate understanding.
Author Response
Dear Sir
Thanks for excellent suggestions. I made changes to accommodate your suggestions. I also sought help from professionals to fix English problems. I send English certificate.
Sincerely yours
Jeoung Dooil
Professor of Biochemistry
Kangwon National University
Chuncheon 24341, Korea
Q1. This review addresses an important topic, the potential of senescence as an anti-cancer therapy. The authors have made a significant effort describing, often very briefly, the conclusions of many publications. However, the result of this is that the narrative is very difficult to follow as not enough background is provided for the reader to understand the specific conclusions. Sometimes, there is just one concluding sentence and the reader wonders whether the experimental design was in cells or in vivo in mice or humans etc. The authors could provide further background and group their conclusions, e.g evidence in vitro, in vivo in mice and humans, or in any other way to compartmentalise the narrative so it is better structured and organised.
Ans. Thanks for excellent suggestions. I make these changes to accommodate the suggestions.
â‘ In this revision, I first try to give more information (background, in vitro/in vivo experimental evidence conclusion) to make readers to better understand this manuscript. In this revision, I try not to give brief conclusions. Please take a look at new manuscript.
â‘¡ I remove unnecessary sentences to make this manuscript more readable.
â‘¢ In this revision, I give more information in each figure.
â‘£ In this revision, I try to give experimental details.
⑤ I also change figures in a way to give more information.
Q2. The focus on autophagy, HDACs and miRNAs is not explained, and the reader wonders why to chose these three subjects. I do not find these particularly interesting or relevant. However, I miss sections discussing the field of senotherapies; senolytics, senomorphics, biotherapeutics, PROTACs, pro-drugs, etc
Ans. Thanks. I understand your concern. I initially thought that inclusion of autophagy, miRNAs, and HDACs can help readers to better understand the mechanisms of senescence. I hope that this does not bother you too much. The below might provide reasons why I add autophagy, miRNAs, and HDACs. I hope that HDAC inhibitors/activators, miRNAs (miR-mimic, miR-inhibitor), and peptides can be developed as senolytic drugs.
â–· I mention the relationship between autophagy and senescence: Please take look at lines 369-374.
[ Radiation-induced senescence involved increased autophagy in lung cancer cells [13]. Senescent cells appear large, flattened, and irregularly shaped, which is attributed to increased mammalian target of rapamycin (mTOR) signaling [108, 109]. Derivatives of indolo [2,3-a] pyrrolo [3,4-c] carbazole induce senescence by modulating AKT/mTOR/S6K signaling in breast cancer cells, lung cancer cells, and colon cancer cells [109]. These reports suggest close relationship between autophagy and senescence].
* SASP regulates anticancer drug resistance (Figure 4) and it is known that autophagy promotes anticancer drug resistance (refs. 115, 117). This also shows the relationship between autophagy and senescence.
â–· I mention the roles of miRNAs in senescence: Please take a look at lines 449-457.
Senescence increases the secretion of EVs, components of the SASP, and EVs can transport proteins and miRNAs. Since the SASP mediates cellular interactions, miRNAs may play critical roles in the spread of senescence. miR-34a targets Wnt and Notch pathways, as well as BCL2, and was found in EVs secreted from subjects with senescence [124]. Thus, targeting miR-34a may improve senescence. miR-485-5p targeted CDK16 to decrease the expression of C-MYC and PD-L1, which, in turn, induced the senescence associated phenotypes in lung cancer cell lines [125]. Therefore, CDK16 can be a target for the induction of senescence in cancer cells. Thus, miR-485-5p mimic could be employed for inducing senescence in cancer cells.
â–· I mention the roles of HDACs in senescence: Please take a look at lines 517-525.
4-Phenylbutyric acid (4-PBA), an HDAC inhibitor (HDACi), induced EMT in gastric cancer cell lines by upregulating IL-8 [135]. IL-8, a component of the SASP, is known to mediate immune suppression and the recruitment of MDSCs (Figure 4). This further suggests the role of HDACs in senescence. SAHA (vorinostat), an HDACi, or the specific downregulation of HDAC2 or HDAC7 by siRNA, induced senescence in dermal fibroblasts by increasing the expression of IL-6, IL-8, MMP-1/-3, and IL-1β [24]. The ectopic overexpression of HDAC7 by lentiviral transduction in pre-senescent dermal fibroblasts extended their proliferative lifespan [24]. These reports suggest that HDACs could inhibit senescence.
â–· I mention peptide as a senescence inducing drug: Please take a look at lines 619-624.
Defensin-like peptide CopA3 increased SA-β-gal activity and ROS levels and induced G1 cell cycle arrest in colon cancer cells [159]. Thus, CopA3 is useful for the induction of senescence in cancer cells. CopA3 can be used with current chemotherapeutic drugs to eliminate senescent cancer cells. The peptides that can increase the expression of p53 or other inducers of senescence may enhance the sensitivity of cancer cells to current anticancer drugs.
â–· I mention senolytic drugs: Please take a look at lines 649-671.
Panobinostat, a Food and Drug Administration (FDA)-approved histone deacetylase inhibitor, eliminated TIS cancer cells (NSCLCs and head and neck squamous cell carcinoma) by decreasing the expression of BCL-XL [164]. Since TIS involves the increased expression of BCL-2, targeting BCL-2 may exert anticancer effects. BCL2-targeting drugs, including ABT-737 and ABT-263 (navitoclax), natural substances, such as artesunate, fisetin, and curcumin, showed senolytic effects in glioblastoma cells and non-transformed cells [75, 165]. ABT-263 eliminated senescent cells and enhanced the efficacy of olaparib, a PARP inhibitor, in both breast and ovarian cancer cell lines [166]. Olaparib increased the expression of Bcl-XL, inflammatory cytokines, check point kinase 2 (Chk2), and p21 [166]. Thus, PARPi and senolytic drugs can induce efficient clinical outcomes. Oridonin, an active ingredient of Chinese herbal medicine, targets glutathione S-transferases to activate ROS-p38 signaling and eliminate senescent cell via apoptosis [167]. Doxorubicin induces senescence in lung cancer cells by increasing cytoplasmic p21 and the senolytic effect of ABT-263 was enhanced in p21 knockout lung cancer cells [168]. Senolytic drugs, such as dasatinib, quercetin, and fisetin, eliminated doxorubicin-induced senescent cells by inducing apoptosis [21]. Doxorubicin increased the expression of the SASP to induce senescence in endothelial cells [21]. Dasatinib and quercetin attenuated cisplatin-induced ovarian injury by targeting senescent cells and reducing DNA damage in primary granulosa cells [169]. Fisetin induced G2/M arrest and apoptosis in NSCLCs by inhibiting EGFR/ERK1/2/STAT3 signaling [170]. Thus, TIS can sensitize cancer cells to senolytic drugs. The identification of the targets of these senolytic drugs may help us to better understand the mechanisms of senescence.
Q3. Overall, I feel that the subject is very interesting but the current text needs extensive revision to improve the structure and background supporting the conclusions. I also find that the authors have cited evidence after evidence without these being assessed; in other words, the authors' vision or critical assessment of the evidence is currently missing. This is usually an important part of a review that helps synthesise the evidence and facilitate understanding.
Ans. I agree. Thanks for excellent suggestions.
â‘ In this revision, I change each figure in a way to give more information. This might help readers better understand this manuscript. Please take a look at new figures.
â‘¡ I remove unnecessary sentences to make this manuscript more readable.
â‘¢ In this revision, I try to include conclusion and assessment for each finding throughout the manuscript. Please take a look at new manuscript.
â‘£ I add/delete several references to make this manuscript more readable.
⑤ I add more references on senolytic drugs: refs. 169, 170. I also add references on peptides as possible senolytic drugs.
â‘¥ In this revision, I try to improve the structure and background to support conclusions.
Reviewer 3 Report
Jo, Shim and Jeoung describe the potential of senescence for cancer therapies.
This is quite an interesting study but there are some points that have to be clarified.
In detail, I have the following suggestions to improve the manuscript:
Chapter „Characteristics of Senescence“
1. Line 24: please add important papers that describe the cancer-protecting effects of senescent cells like:
· He and Sharpless, Senescence in Health and Disease, Cell, 2017
· Campisi, Aging, cellular senescence and cancer, Annu Rev Physiol, 2013
· Braumüller et al., T-helper-1-cell cytokines drive cancer into senescence, Nature, 2013
1. Line 26: please add references
2. Figure 1: In this figure, you show the loss of mitochondria in senescent cells but you do not mention this in the text. The authors should say a sentence about this astonishing and new effect and add a reference.
Chapter “Factors inducing Senescence”
1. I recommend that the authors use separate paragraphs for the different factors like
(I) Chemotherapies and Radiotherapies
(II) Cell cycle Inhibitors
(III) Epigenetic Modulators
This would make the paragraph much easier to read.
2. Line 46: In human cancer cells, low doses of chemotherapy trigger senescence while higher doses trigger apoptosis. I suggest that the authors say one or two sentences about this effect.
3. Line 50: Doxorubicin, etoposide and camptothecin are topoisomerase inhibitors that lead to massive DNA damage and increased expression of p53, CDKN1A and SERPINE1. Please explain the effects of topoisomerase inhibitors on senescence.
4. Please say one or two sentences about platinum-based drugs and why these drugs induce increased expression of p53.
5. Please explain why chemotherapies and radiotherapies that induce DNA damage and subsequently p53 upregulation induce senescence although p53 is known as regulator of apoptosis.
6. Figure 2: this figure is very busy and therefore difficult to understand. I suggest that the authors separate the figure into a figure that shows the stimuli and mechanisms of senescence induction and the therapy-induced senescence induction.
Chapter “Oncogene-induced Senescence”
1. Line 97: As the authors describe that anti-cancer therapies like chemotherapeutic drugs or radiotherapies induce DNA damage and ROS and do not target oncogenes, this sentence does not make sense. Please remove it.
2. Line 100: Please explain the contradictory findings that senescent cancer cells, although permanently growth arrested as described in line 22, become more aggressive.
3. Line 102: immune checkpoint inhibitor therapy tries to activate immune cells. Why do cancer cells like TNBCs react or even get resistant after antibody injection? Please explain.
4. Line 105: I do not understand these sentences. The authors write that targeting oncogenes leads to the elimination of senescent cells. Like treatment with senolytic agents? In addition, why should this prevent transformation of senescent cells? Please explain this.
5. The authors show that most OIS induction is p53 dependent. In human cancer cells TP53 and RB1 pathways are frequently mutated although, most cancercell lines can be rendered senescent in vitro by e.g., AURK inhibitors. Is there an explanation for this?
6. Line 153: The authors write in line 100 that senescent cells are permanently growth arrested. In line 145, the authors write that senescent cells transform into cancer cells. Is a cell that will never divide a cancer cell anymore?
7. To avoid confusion about the point that a senescent cell is permanently growth arrested and can transform into a cancer cell the authors should add a paragraph about the stability of senescence and how senescence is maintained.
Chapter “Roles of SASP in Cancer Cell Proliferation and Therapeutic Resistance”
1. The authors should explain that the SASP differs according to the senescence inducing stimuli. Judith Campisi and collaborators were the first that described the different SASP factors 2016.
2. Line 172: reference 57 is a study in zebrafish and not in humans.
3. Line 178: what kind of cells form this tumor-like cell mass? Please explain.
4. Line 216: the authors explain the resistance of senescent cells to anticancer drugs with the upregulation of anti-apoptotic proteins. Most chemotherapeutic drugs need highly proliferating cells to be effective and senescent cells are growth arrested, this could be another explanation for the resistance. Please discuss other explanations.
5. Line 215: With this very interesting paper, it is the same problem as before. Permanently arrested cells became proliferating cancer cells. I recommend that the authors insert a paragraph that describes the stability of senescence.
6. Line 220: targeting BCL-2 or other anti-apoptotic proteins is known as senolysis or senolytic therapy.
7. Line 212: I have never heard that EGFR and HER2 tyrosine kinases inhibitors like lepatinib and fulvestran can induce senescence. Please explain this astonishing effect.
8. Line 224: inflammatory endothelial cells secrete high levels of IL-8 without being senescent. Targeting IL-8 can of course improve clinical outcome but not necessary as part of the SASP.
Chapter “Targeting Therapy-induced senescence”
1. In my opinion, this paragraph is not necessary because targeting TIS is already described in the previous paragraphs and the MDM2-p53 disruptors like nutlin 3 can be added in the “Factors inducing Senescence” paragraph.
Chapter “Role of Autophagy in Senescence”
1. Line 359: the authors write that it is still unclear whether autophagy has a positive or negative effect on senescence but in the whole paragraph they only cite references that show a positive effect of autophagy on cancer cell senescence. Please add examples for negative effects of autophagy on senescence.
2. Line 375: just because both autophagy and senescence induce anticancer drug resistance, one cannot say that CAGE induces senescence. To relate autophagy and senescence with the reference the authors cite is very speculative.
3. Line 408: is this sentence true for all anti-cancer drugs and all senescence-inducing stimuli?
Chapter “Role of HDACs in Senescence”
1. This paragraph is very inconsistent, it describes HDACs and apoptosis and senescence and histone methylation. I recommend to rewrite it and only describe HDACs and senescence.
2. Line 476: please explain what HDACs are and why they are important epigenetic regulators.
3. Line 478: the authors describe the methylation of histones in the HDAC paragraph. Please do not mingle acetylation and methylation.
4. Line 484: not every cell that secretes a pro-inflammatory cytokine like IL-8 is senescent. Please specify the markers for senescence used in this study.
5. Line 494: caspase 3/7 is no senescence marker but a marker for apoptosis. The inhibition of HDAC2/HDAC6 together with cisplatin seems to enhance apoptosis and not senescence.
6. Line 496: the authors want to discuss the role of HDACs in senescence and not the role of HDACs in apoptosis.
Chapter “Conclusion and Perspective”
1. Line 574-579: In the first sentence, the authors say that senescent cells can acquire an aggressive phenotype and in the second sentence, the senescent tumour cell is permanently growth arrested. This contradiction must be explained.
2. Line 581: How does CopA3 induce senescence? Like a CDK4/6 inhibitor?
3. Line 585-585: in the first sentence the authors explain that factors of the SASP of senescent cells can induce the upregulation of PD-L1. In the third sentence the authors write that upregulation of PD-L1 on non-senescent cells by chemotherapeutic drugs induces senescence. Is the upregulation of a checkpoint inhibitor molecule enough to induce senescence? In the previous paragraphs the authors explain that chemotherapeutic drugs induce senescence by DNA damage. Please explain these contradictory findings.
4. Line 590: I do not understand this sentence. More PD-L1 on senescent cancer cells should enhance the efficiency of PD-1 antibodies that target PD-1 on immune cells? How and why?
5. Line 605: the main task of senolytic drugs is the elimination of senescence cells by targeting anti-apoptotic pathways!
6. Line 609: usually cells upregulate more than one ant-apoptotic proteins so targeting only the BCL-pathway should result in apoptosis induction in a restricted range of senescent cell types.
7. Line 640: despite the Silybum marianum flower extract the authors mention there are more SASP inhibitors like Rapamycin or the anti-diabetic drug metformin. Please discuss some more SASP inhibitors.
8. Line 649: please specify the peptide.
9. Please describe the advantages and disadvantages of senolytics versus SASP inhibitors.

Author Response
Dear Sir
Thanks for excellent suggestions. I made changes according to the suggestions.
I hope that changes I made are fine.
Sincerely yours
Jeoung Dooil, Ph.D.
Professor of Biochemistry
Kangwon National University
Chuncheon 24341, Korea
Chapter “Characteristics of Senescence”
Q.Line 24: please add important papers that describe the cancer-protecting effects of senescent cells like:
He and Sharpless, Senescence in Health and Disease, Cell, 2017
Campisi, Aging, cellular senescence and cancer, Annu Rev Physiol, 2013
Braumüller et al., T-helper-1-cell cytokines drive cancer into senescence, Nature, 2013
Ans. I add the above three references (refs. 6-8). Please take look at new references.
Q1. Line 26: please add references
Ans. I add reference (ref.9). Please take look at new references.
Bousset L, Gil J. Targeting senescence as an anticancer therapy. Mol Oncol. 2022 Nov;16(21):3855-3880. doi: 10.1002/1878-0261.13312.
Q2. Figure 1: In this figure, you show the loss of mitochondria in senescent cells but you do not mention this in the text. The authors should say a sentence about this astonishing and new effect and add a reference.
Martini H, Passos JF. Cellular senescence: all roads lead to mitochondria. FEBS J. 2022 Jan 20:10.1111/febs.16361. doi: 10.1111/febs.16361. Online ahead of print.
Ans. I add the above refence. This reference shows the effect of mitochondrial dynamics on senescence. Please take look at new references.
I add this sentence: Senescence is also characterized by changes in mitochondria dynamics, structure, and function [17]. Impaired mitochondrial dynamics lead to senescence [17]. Please take look at lines 37-38.
Chapter “Factors inducing Senescence”
Q1.I recommend that the authors use separate paragraphs for the different factors like
(I) Chemotherapies and Radiotherapies
(II) Cell cycle Inhibitors
(III) Epigenetic Modulators
This would make the paragraph much easier to read.
Ans. Thanks. I rearrange order of paragraphs and sentences. I remove unnecessary sentences to make it easier to read. Please take look at new manuscript.
Q2. Line 46: In human cancer cells, low doses of chemotherapy trigger senescence while higher doses trigger apoptosis. I suggest that the authors say one or two sentences about this effect.
Ans. Thanks. I add the sentence: Low doses of chemotherapy trigger senescence while higher doses trigger apoptosis. Please take look at lines 57-58. It is probable that higher doses may cause complete autophagy while low doses may cause impaired autophagy.
Q3. Line 50: Doxorubicin, etoposide and camptothecin are topoisomerase inhibitors that lead to massive DNA damage and increased expression of p53, CDKN1A and SERPINE1. Please explain the effects of topoisomerase inhibitors on senescence.
Odeh A, Dronina M, Domankevich V, Shams I, Manov I. Downregulation of the inflammatory network in senescent fibroblasts and aging tissues of the long-lived and cancer-resistant subterranean wild rodent, Spalax. Aging Cell. 2020 Jan;19(1):e13045. doi: 10.1111/acel.13045.
Ans. I add the sentence: Etoposide, an inhibitor of topoisomerase, induced senescence by increasing the expression of p53, p21, and p16 in Spalax fibroblasts. Please take look at lines 62-64. I add the above reference. The above reference shows the effect of etoposide on senescence and p53 expression. Please take look at new references.
Q4. Please say one or two sentences about platinum-based drugs and why these drugs induce increased expression of p53.
Ans. I add this sentence: Topoisomerase inhibitors, such as, doxorubicin and etoposide, may increase the expression of p53 by inducing DNA damage. Please take look at lines 67-69.
Q5.Please explain why chemotherapies and radiotherapies that induce DNA damage and subsequently p53 upregulation induce senescence although p53 is known as regulator of apoptosis.
Ans. P53 displays diverse roles, including apoptosis and senescence. P53 activation in response to DNA damage causes growth arrest, allowing for DNA repair, or directs senescence or apoptosis. The induction of p53 is necessary for DNA repair. For DNA repair, senescence, but not apoptosis, is induced by p53. P53 increases the expression of p21 and p16, which in turn causes cell cycle arrest to induce senescence. It is probable that low doses of chemotherapy drugs cause senescence via p53 while higher doses of chemotherapy drugs cause apoptosis via p53.
Q6.Figure 2: this figure is very busy and therefore difficult to understand. I suggest that the authors separate the figure into a figure that shows the stimuli and mechanisms of senescence induction and the therapy-induced senescence induction.
Ans. I agree. I separate figure 2 into two parts. Please take look at new figure 2.
Chapter “Oncogene-induced Senescence”
Q1. Line 97: As the authors describe that anti-cancer therapies like chemotherapeutic drugs or radiotherapies induce DNA damage and ROS and do not target oncogenes, this sentence does not make sense. Please remove it.
Ans. I agree. I remove it. Please take look at new manuscript.
Q2.Line 100: Please explain the contradictory findings that senescent cancer cells, although permanently growth arrested as described in line 22, become more aggressive.
Ans. Oncogenes can promote senescence by inducing DNA damage and ROS production. Not all senescent cells remain permanently arrested. Some senescent cells remain active and secrete inflammatory cytokines/chemokines for communications with nearby cells. These cellular communications may allow senescent cells to acquire aggressive phenotypes.
Q3. Line 102: immune checkpoint inhibitor therapy tries to activate immune cells. Why do cancer cells like TNBCs react or even get resistant after antibody injection? Please explain.
Ans. Immune check point inhibitor (ICI), such as anti-PD-L1, can bind to PD-L1 expressed on TNBCs. This binding suppresses the binding of PD-L1 to PD-1 to inhibit immune evasion exerted by cancer cells. Primary resistance mechanisms (to ICI) include insufficient tumor immunogenicity, dysfunction of MHCs, irreversible T cell exhaustion, primary resistance to IFN-γ signaling, and immunosuppressive microenvironment. Some oncogenic signaling pathways also contribute to the primary resistance. Under the pressure applied by anti-PD1/PDL1 therapy, tumors experience immunoediting and preserve beneficial mutations, upregulate the compensatory inhibitory signaling and induce re-exhaustion of T cells, all of which may attenuate the durability of the therapy.
Q4. Line 105: I do not understand these sentences. The authors write that targeting oncogenes leads to the elimination of senescent cells. Like treatment with senolytic agents? In addition, why should this prevent transformation of senescent cells? Please explain this.
Ans. It is known that senolytic drugs target senescent cancer cells. Oncogenes can induce senescence. Some transformed senescent cells can remain active and acquire aggressive phenotypes by secreting inflammatory cytokines/chemokines to induce cellular interactions. Therefore, targeting oncogenes can eliminate these senescent cancer cells by preventing them from inducing cellular interactions for acquiring malignant phenotypes.
Q5. The authors show that most OIS induction is p53 dependent. In human cancer cells TP53 and RB1 pathways are frequently mutated although, most cancercell lines can be rendered senescent in vitro by e.g., AURK inhibitors. Is there an explanation for this?
Ans. It has been known that aurora kinase inhibitors induce the expression of p53 and/or senescence:
- Liu Y, Hawkins OE, Su Y, Vilgelm AE, Sobolik T, Thu YM, Kantrow S, Splittgerber RC, Short S, Amiri KI, Ecsedy JA, Sosman JA, Kelley MC, Richmond A. Targeting aurora kinases limits tumour growth through DNA damage-mediated senescence and blockade of NF-κB impairs this drug-induced senescence. EMBO Mol Med. 2013 Jan;5(1):149-66. doi: 10.1002/emmm.201201378.
- Liu X, Shi Q, Choudhry N, Zhang T, Liu H, Zhang S, Zhang J, Yang D. The Effect of Circumscribed Exposure to the Pan-Aurora Kinase Inhibitor VX-680 on Proliferating Euploid Cells. Int J Mol Sci. 2022 Oct 11;23(20):12104. doi: 10.3390/ijms232012104.
Since AURK inhibitors induce the expression of p53, these inhibitors can induce senescence.
Q6. Line 153: The authors write in line 100 that senescent cells are permanently growth arrested. In line 145, the authors write that senescent cells transform into cancer cells. Is a cell that will never divide a cancer cell anymore?
Ans. Senescent cells are unique in that they eventually stop multiplying but don’t die off when they should. Not all senescent cells remain permanently arrested. Some senescent cells remain active and secrete inflammatory cytokines/chemokines for communications with nearby cells. These cellular interactions enable senescent cells to acquire aggressive phenotypes.
Q7.To avoid confusion about the point that a senescent cell is permanently growth arrested and can transform into a cancer cell the authors should add a paragraph about the stability of senescence and how senescence is maintained.
Ans. Thanks. I agree. I add this sentence: Not all senescent cells remain permanently arrested. Some senescent cells remain active and secrete inflammatory cytokines/chemokines for communications with nearby cells. These cellular communications may allow senescent cells to acquire aggressive phenotypes. Please take look at lines 124-127.
Chapter “Roles of SASP in Cancer Cell Proliferation and Therapeutic Resistance”
Q1. The authors should explain that the SASP differs according to the senescence inducing stimuli. Judith Campisi and collaborators were the first that described the different SASP factors 2016.
Ans. Thanks. I add this sentence: Composition of SASP shows variation depending on the stimuli, the cell type, and the stage of senescence. Please take look at lines 671-672. I add references below.
â‘ Wiley CD, Velarde MC, Lecot P, Liu S, Sarnoski EA, Freund A, Shirakawa K, Lim HW, Davis SS, Ramanathan A, et al. Mitochondrial Dysfunction Induces Senescence with a Distinct Secretory Phenotype. Cell Metab. 2016 23,303-14. doi: 10.1016/j.cmet.2015.11.011.
â‘¡Wiley CD, Flynn JM, Morrissey C, Lebofsky R, Shuga J, Dong X, Unger MA, Vijg J, Melov S, Campisi J. Analysis of individual cells identifies cell-to-cell variability following induction of cellular senescence. Aging Cell 2017, 16, 1043-1050. doi: 10.1111/acel.12632.
Q2. Line 172: reference 57 is a study in zebrafish and not in humans.
Ans. I change. Thanks. Please take look at line 201.
Q3. Line 178: what kind of cells form this tumor-like cell mass? Please explain.
Ans. Tumor-like cell mass include transformed RASG12V mutant cells that became senescent and eliminated, RASG12V-TP53R175H double-mutant cells, and normal cells. The surviving double mutant cells secrete SASP-like inflammatory cytokines to convert nearby normal cells into SASP factor-secreting senescent cells.
Q4. Line 216: the authors explain the resistance of senescent cells to anticancer drugs with the upregulation of anti-apoptotic proteins. Most chemotherapeutic drugs need highly proliferating cells to be effective and senescent cells are growth arrested, this could be another explanation for the resistance. Please discuss other explanations.
Ans. Senescence plays paradoxical roles in cancer. Induction of senescence inhibits cancer progression and enhances the sensitivity to anticancer drug but lingering (metabolically active) senescent cells fuel progression, recurrence, and metastasis. Lingering senescent cells are metabolically active and can display resistance to anticancer drugs. These lingering senescent cells secrete SASP to affect nearby cells. Cellular interactions driven by SASP may affect phenotypes of these senescent cells.
Q5. Line 215: With this very interesting paper, it is the same problem as before. Permanently arrested cells became proliferating cancer cells. I recommend that the authors insert a paragraph that describes the stability of senescence.
Ans. I agree. I add these sentences: Senescent cells are unique in that they eventually stop multiplying but don’t die off when they should. A relatively small number of senescent cells can remain active and spread senescence (lines 39-41). The irreversible senescence is mediated by INK4a/Rb and p53 pathways and the reversible senescent phenotype is mediated by p53. This suggests that p53 pathway could be effectively harnessed as a therapeutic intervention to trigger senescence and ultimately suppress tumorigenesis (lines 71-74). Senescent cells can undergo conversion to an immunogenic phenotype that enables them to be eliminated by the immune system (lines 327-328). I hope that these sentences may give partial answers.
Q6. Line 220: targeting BCL-2 or other anti-apoptotic proteins is known as senolysis or senolytic therapy.
Ans. Thanks. I agree. I add this sentence: Thus, targeting BCL-2 or other anti-apoptotic proteins can eliminate senescent cancer cells (senolysis or senolytic therapy). Please take look at lines 254-256.
Q7. Line 212: I have never heard that EGFR and HER2 tyrosine kinases inhibitors like lepatinib and fulvestran can induce senescence. Please explain this astonishing effect.
Ans. Thanks. I agree. I change the sentence into: Lapatinib and fulvestrant (EGFR and HER2 inhibitors) treatment induced apoptosis in chemotherapy-induced senescent breast cancer cells [83]. Please take look at lines 251-252.
Q8. Line 224: inflammatory endothelial cells secrete high levels of IL-8 without being senescent. Targeting IL-8 can of course improve clinical outcome but not necessary as part of the SASP.
Ans. Thanks. I agree. I thank your outstanding perspectives. I just wanted to say some possible role of IL-8 in anticancer drug resistance. IL-8 can be a component of SASP in a context-dependent manner.
Chapter “Targeting Therapy-induced senescence”
- In my opinion, this paragraph is not necessary because targeting TIS is already described in the previous paragraphs and the MDM2-p53 disruptors like nutlin 3 can be added in the “Factors inducing Senescence” paragraph.
Ans. Thanks. I agree. I delete the paragraph. Please take look at new manuscript. I put reference concerning MDM2-p53 disruptors like nutlin 3 into “Factors inducing Senescence” paragraph. Reference 124 is now ref 24.
Chapter “Role of Autophagy in Senescence”
Q1. Line 359: the authors write that it is still unclear whether autophagy has a positive or negative effect on senescence but in the whole paragraph they only cite references that show a positive effect of autophagy on cancer cell senescence. Please add examples for negative effects of autophagy on senescence.
Ans. I add this reference: Fan X, He Y, Wu G, Chen H, Cheng X, Zhan Y, An C, Chen T, Wang X. Sirt3 activates autophagy to prevent DOX-induced senescence by inactivating PI3K/AKT/mTOR pathway in A549 cells. Biochim Biophys Acta Mol Cell Res. 2023,1870, 119411. doi: 10.1016/j.bbamcr.2022.119411. This shows a negative regulatory role of autophagy in senescence.
Please take look at lines 360-361 and new reference (ref. 114).
Q2.Line 375: just because both autophagy and senescence induce anticancer drug resistance, one cannot say that CAGE induces senescence. To relate autophagy and senescence with the reference the authors cite is very speculative.
Ans. I agree. I rather delete this sentence: Since senescence can induce anticancer drug resistance, it is probable that CAGE may also induce senescence to promote anticancer drug resistance. Please see new manuscript.
Q3. Line 408: is this sentence true for all anti-cancer drugs and all senescence-inducing stimuli?
Ans. I think that it may not be true for all anti-cancer drugs. I rather delete the sentence: Thus, targeting autophagy may enhance the effects of anticancer drugs by suppressing senescence. Please take look at new manuscript.
Chapter “Role of HDACs in Senescence”
Q1. This paragraph is very inconsistent, it describes HDACs and apoptosis and senescence and histone methylation. I recommend to rewrite it and only describe HDACs and senescence.
Ans. I agree. I remove unnecessary sentences and references. I try to mention only HDACs and senescence. Please take look at new manuscript. New paragraph focuses on HDACs and senescence.
Q2. Line 476: please explain what HDACs are and why they are important epigenetic regulators.
Ans. It is well known that histone modifications regulate gene expression. Tet methylcytosine dioxygenase 2 (TET2) inhibited PD-L1 gene expression by recruiting HDACs in MDA-MB-231 cells [136].
They are HDAC1 and -2. TET2 recruits HDAC1/2 to deacetylate H3K27ac at PD-L1 promoter. Histone deacetylation at PD-L1 promoter decreases the expression of PD-L1. It is known that PD-L1/PD-1 axis is necessary for senescence. Thus, immune checkpoint inhibitor (ICI) therapy can promote senescence surveillance which eliminates senescent cancer cells. The reference below shows the effect of ICI on senescence surveillance. It is thus reasonable that HDAC1/2 may induce immune surveillance (senescence surveillance) by decreasing the expression of PD-L1.
Wang TW, Johmura Y, Suzuki N, Omori S, Migita T, Yamaguchi K, Hatakeyama S, Yamazaki S, Shimizu E, Imoto S, Furukawa Y, Yoshimura A, Nakanishi M. Blocking PD-L1-PD-1 improves senescence surveillance and ageing phenotypes. Nature. 2022 Nov;611(7935):358-364. doi: 10.1038/s41586-022-05388-4.
Q3. Line 478: the authors describe the methylation of histones in the HDAC paragraph. Please do not mingle acetylation and methylation.
Ans. I agree. I understand. I just wanted to show that histone methylation, just like histone acetylation/deacetylation, can affect senescence. In other words, histone modifications generally could affect senescence. I do not intend to mingle acetylation and methylation. I am sorry to cause confusion.
Q4. Line 484: not every cell that secretes a pro-inflammatory cytokine like IL-8 is senescent. Please specify the markers for senescence used in this study.
Ans. This study does not show the effect of 4-Phenylbutyric acid (4-PBA), an HDAC inhibitor (HDACi) on senescence. However, 4-PBA decreased the expression of IL-8, a component of SASP. I just wanted to show a possible role of HDACs in senescence. I agree that this study does not show direct relationship between HDACs and senescence. I am sorry to cause confusion.
Q5. Line 494: caspase 3/7 is no senescence marker but a marker for apoptosis. The inhibition of HDAC2/HDAC6 together with cisplatin seems to enhance apoptosis and not senescence.
Ans. I agree. I delete the sentence and the reference. Please take look at new manuscript.
Q6.Line 496: the authors want to discuss the role of HDACs in senescence and not the role of HDACs in apoptosis.
Ans. I agree. I delete the sentence and the reference. Please take look at mew manuscript.
Chapter “Conclusion and Perspective”
Q1. Line 574-579: In the first sentence, the authors say that senescent cells can acquire an aggressive phenotype and in the second sentence, the senescent tumour cell is permanently growth arrested. This contradiction must be explained.
Ans. I agree. It is my mistake. Senescent cells are unique in that they eventually stop multiplying, but not die off. I did not mean that all senescent cells are permanently growth arrested. All senescent cells do not stay permanently arrested. Some senescent cells remain metabolically active and secrete pro-inflammatory cytokines. A small number of senescent cells remain and release chemicals that can cause inflammation in nearby cells. These cellular interactions enable senescent cells to acquire aggressive phenotypes. Please take look at lines 39-41.
Q2. Line 581: How does CopA3 induce senescence? Like a CDK4/6 inhibitor?
Ans. Defensin-like peptide CopA3 increased SA-β-gal activity and ROS levels and DNA damage and G1 cell cycle arrest in colon cancer cells. It is not clear whether this peptide display CDK inhibition effect. Since it induces G1 arrest and senescence, it is possible that CopA3 peptide can decrease CDK activity. It is well known that CDK is necessary for cell cycle progression.
Q3.Line 585-585: in the first sentence the authors explain that factors of the SASP of senescent cells can induce the upregulation of PD-L1. In the third sentence the authors write that upregulation of PD-L1 on non-senescent cells by chemotherapeutic drugs induces senescence. Is the upregulation of a checkpoint inhibitor molecule enough to induce senescence? In the previous paragraphs the authors explain that chemotherapeutic drugs induce senescence by DNA damage. Please explain these contradictory findings.
Ans. The references below show that doxorubicin increases the expression of PD-L1. The increased expression of PD-L1 causes immune evasion. PD-L1/PD-1 axis has been known to be involved in senescence. The blocking of PD-L1/PD-1 axis induces senescence surveillance (elimination of senescent cells). I would not say upregulation of PD-L1 alone can induce senescence. However, upregulation of PD-L1, at least partially, is necessary for the induction of senescence by chemotherapy. Thus, chemotherapy drugs not only induce DNA damage but also induce PD-L1 expression.
â‘ Wang J, Hu F, Yu P, Wang J, Liu Z, Bao Q, Zhang W, Wen J.J. Sorafenib inhibits doxorubicin-induced PD-L1 upregulation to improve immunosuppressive microenvironment in Osteosarcoma. Cancer Res Clin Oncol. 2022 Nov 8. doi: 10.1007/s00432-022-04458-4. Online ahead of print.
â‘¡Yati S, Silathapanasakul A, Thakaeng C, Chanasakulniyom M, Songtawee N, Porntadavity S, Pothacharoen P, Pruksakorn D, Kongtawelert P, Yenchitsomanus PT, Chanmee T. Extracellular Vesicle-Mediated IL-1 Signaling in Response to Doxorubicin Activates PD-L1 Expression in Osteosarcoma Models. Cells. 2022 Mar 18;11(6):1042. doi: 10.3390/cells11061042.
â‘¢Wang TW, Johmura Y, Suzuki N, Omori S, Migita T, Yamaguchi K, Hatakeyama S, Yamazaki S, Shimizu E, Imoto S, Furukawa Y, Yoshimura A, Nakanishi M. Blocking PD-L1-PD-1 improves senescence surveillance and ageing phenotypes. Nature. 2022 Nov;611(7935):358-364. doi: 10.1038/s41586-022-05388-4.
Q4. Line 590: I do not understand this sentence. More PD-L1 on senescent cancer cells should enhance the efficiency of PD-1 antibodies that target PD-1 on immune cells? How and why?
Ans. For example, anti-PD-L1 antibody can bind to PD-L1 on senescent cancer cells. The binding of anti-PD-L1 antibody to PD-L1 inhibits the binding of PD-L1 to PD-1 to suppress immune evasion. Therefore, the increased expression of PD-L1 on senescent cancer cells enhances the sensitivity of cancer cells to PD-L1/PD-1 blockade.
Q5. Line 605: the main task of senolytic drugs is the elimination of senescence cells by targeting anti-apoptotic pathways!
Ans. I add the sentence: the main task of senolytic drugs is the elimination of senescence cells by targeting anti-apoptotic pathways. Please take look at lines 614-615.
Please also take look at lines 620-622:BCL2-targeting drugs, including ABT-737 and ABT-263 (navitoclax), natural substances, such as artesunate, fisetin, and curcumin, showed senolytic effects in glioblastoma cells and non-transformed cells [82,166].
Q6.Line 609: usually cells upregulate more than one ant-apoptotic proteins so targeting only the BCL-pathway should result in apoptosis induction in a restricted range of senescent cell types.
Ans. Thanks. I agree. I add this reference. Please take look at lines. 617-619. This reference shows Wnt/β-catenin signaling as another target of senolytic drugs.
Qin G, Li Y, Xu X, Wang X, Zhang K, Tang Y, Qiu H, Shi D, Zhang C, Long Q, et al. Panobinostat (LBH589) inhibits Wnt/β-catenin signaling pathway via upregulating APCL expression in breast cancer. Cell Signal. 2019 Jul;59:62-75. doi: 10.1016/j.cellsig.2019.03.014.
Q7.Line 640: despite the Silybum marianum flower extract the authors mention there are more SASP inhibitors like Rapamycin or the anti-diabetic drug metformin. Please discuss some more SASP inhibitors.
Ans. I add reference below (rapamycin). I also add this sentence: Rapmycin, an inducer of autophagy, inhibits senescence and SASP by decreasing SA-β-Gal activity and p16 expression in human coronary artery endothelial cells (ref 174). Please take look at lines 664-666.
Sasaki N, Itakura Y, Toyoda M. Rapamycin promotes endothelial-mesenchymal transition during stress-induced premature senescence through the activation of autophagy.
Cell Commun Signal. 2020 Mar 12;18(1):43. doi: 10.1186/s12964-020-00533-w.
Q8. Line 649: please specify the peptide.
Ans. Thanks. It is defensin-like CopA3 peptide. It is already mentioned. I rather delete the sentence.
Q9. Please describe the advantages and disadvantages of senolytics versus SASP inhibitors.
I mentioned them in the manuscript. Please take look at lines (640-645) and (667-672).
Ans.
< Senolytics >
Senescent cells are like cancer cells that do not divide, including metabolic shift, epigenetic change, and resistance to apoptosis. Senescent cells increase several networks of anti-apoptotic regulators, PI3K/Akt pathway components, and BCL-2 family members, which collectively confer resistance to apoptosis. First generation of senolytics inhibit a portion of these pathways and induce apoptosis preferentially in senescent cells. There are two critical considerations for developing and applying senolytics: first, it is necessary to target a whole pathway for survival of senescent cells, not a single gene in that pathway as other components in the same pathway can compensate for its loss. Second, aiming for one survival pathway alone is not sufficient to eliminate all different populations of senescent cells as they are quite heterogenous.
The current version of senolytics has its own limitation. Since most senolytic drugs were repositioned to target aging and age-related diseases, they might have undesirable side effects for long-term use. Therefore, it is required to reevaluate the safety of a senolytic drug for long-term use if it is originally designed for short-term use. Another potential problem is tissue atrophy, resulting from a massive removal of senescent cells by senolytic drugs. Senescent cells play a role to support the structure of old tissues and thus their abrupt removal may lead to atrophy, depending on the levels of senescent cells that reside in the tissue (He and Sharpless, 2017). In addition, senolytics wipes out not only deleterious effects of senescent cells but also their beneficial effects; cellular senescence contributes to wound healing (Demaria et al., 2014), cellular reprogramming (Mosteiro et al., 2016), and tissue regeneration (Ritschka et al., 2017), suggesting that complete elimination of senescent cells may hamper these beneficial effects of senescent cells.
<SASP inhibitors>
For generalized senostatics, several theraputic targets have been recently emerged as the SASP regulatory network: NF-κB, p38, GATA4, mTOR, BRD4, and cGAS/STING. One potential problem for targeting such regulatory network is that they have also non-senescence related functions; NF-κB plays an essential role in controlling acute inflammatory response and immune response (Chien et al., 2011); p38 is involved in several stress responses including DNA damage, heat shock, and osmotic shock responses (Freund et al., 2011); GATA4 plays a major role in controlling embryonic development (Molkentin et al., 1997); mTOR governs cell growth, proliferation, protein synthesis, and autophagy (Herranz et al., 2015; Laberge et al., 2015); BRD4 functions as an epigenetic reader to modulate lineage- and cellular state-specific transcription (Tasdemir et al., 2016); finally, cGAS-STING is a component of the innate immune system that senses the presence of cytosolic DNA (Yang et al., 2017). Therefore, care must be considered to tailor therapeutic strategies to exclusively blunt the SASP regulation but no other functions. Precision senostatics have not been seriously considered yet mainly because the composition of the SASP highly varies, depending on the cell type, the stage of senescence (early versus late senescence), and the type of senescence-inducing stimuli. Moreover, the composition of the SASP is quite complex, having up to a slightly less than hundred factors. Future studies need to categorize SASP factors according to their functions in a context-dependent manner.
Round 2
Reviewer 1 Report
I critically evaluated the revised version of this manuscript. I found author’s replies are justified and they made the necessary changes as suggested. I believe the present version of the manuscript is suitable for publication.
Author Response
Thanks for your kindness.
Reviewer 2 Report
Thanks for addressing the comments.
Author Response
Thanks for your kindness.
Reviewer 3 Report
Dear Madam,
thank you very much for rewriting the manuscript. In my opinion, this has improved the manuscript.
Still, there are some minor questions I would like you to clarify (marked in red).
Chapter “Characteristics of Senescence”
Q.Line 24: please add important papers that describe the cancer-protecting effects of senescent cells like:
He and Sharpless, Senescence in Health and Disease, Cell, 2017
Campisi, Aging, cellular senescence and cancer, Annu Rev Physiol, 2013
Braumüller et al., T-helper-1-cell cytokines drive cancer into senescence, Nature, 2013
Ans. I add the above three references (refs. 6-8). Please take look at new references.
Q1. Line 26: please add references
Ans. I add reference (ref.9). Please take look at new references.
Bousset L, Gil J. Targeting senescence as an anticancer therapy. Mol Oncol. 2022 Nov;16(21):3855-3880. doi: 10.1002/1878-0261.13312.
Q2. Figure 1: In this figure, you show the loss of mitochondria in senescent cells but you do not mention this in the text. The authors should say a sentence about this astonishing and new effect and add a reference.
Martini H, Passos JF. Cellular senescence: all roads lead to mitochondria. FEBS J. 2022 Jan 20:10.1111/febs.16361. doi: 10.1111/febs.16361. Online ahead of print.
Ans. I add the above refence. This reference shows the effect of mitochondrial dynamics on senescence. Please take look at new references.
I add this sentence: Senescence is also characterized by changes in mitochondria dynamics, structure, and function [17]. Impaired mitochondrial dynamics lead to senescence [17]. Please take look at lines 37-38.
Chapter “Factors inducing Senescence”
Q1.I recommend that the authors use separate paragraphs for the different factors like
(I) Chemotherapies and Radiotherapies
(II) Cell cycle Inhibitors
(III) Epigenetic Modulators
This would make the paragraph much easier to read.
Ans. Thanks. I rearrange order of paragraphs and sentences. I remove unnecessary sentences to make it easier to read. Please take look at new manuscript.
Q2. Line 46: In human cancer cells, low doses of chemotherapy trigger senescence while higher doses trigger apoptosis. I suggest that the authors say one or two sentences about this effect.
Ans. Thanks. I add the sentence: Low doses of chemotherapy trigger senescence while higher doses trigger apoptosis. Please take look at lines 57-58. It is probable that higher doses may cause complete autophagy while low doses may cause impaired autophagy.
Q3. Line 50: Doxorubicin, etoposide and camptothecin are topoisomerase inhibitors that lead to massive DNA damage and increased expression of p53, CDKN1A and SERPINE1. Please explain the effects of topoisomerase inhibitors on senescence.
Odeh A, Dronina M, Domankevich V, Shams I, Manov I. Downregulation of the inflammatory network in senescent fibroblasts and aging tissues of the long-lived and cancer-resistant subterranean wild rodent, Spalax. Aging Cell. 2020 Jan;19(1):e13045. doi: 10.1111/acel.13045.
Ans. I add the sentence: Etoposide, an inhibitor of topoisomerase, induced senescence by increasing the expression of p53, p21, and p16 in Spalax fibroblasts. Please take look at lines 62-64. I add the above reference. The above reference shows the effect of etoposide on senescence and p53 expression. Please take look at new references.
Subterranean wild rodents are quite an exotic animal model. Is there no study in human cancer cell lines or human specimens? At least Etoposide is a chemotherapeutic drug applied in the clinic.
Q4. Please say one or two sentences about platinum-based drugs and why these drugs induce increased expression of p53.
Ans. I add this sentence: Topoisomerase inhibitors, such as, doxorubicin and etoposide, may increase the expression of p53 by inducing DNA damage. Please take look at lines 67-69.
My question was about platinum-based drugs and not topoisomerase inhibitors! Please write one or two sentences to platinum-based chemotherapeutic drugs.
Q5.Please explain why chemotherapies and radiotherapies that induce DNA damage and subsequently p53 upregulation induce senescence although p53 is known as regulator of apoptosis.
Ans. P53 displays diverse roles, including apoptosis and senescence. P53 activation in response to DNA damage causes growth arrest, allowing for DNA repair, or directs senescence or apoptosis. The induction of p53 is necessary for DNA repair. For DNA repair, senescence, but not apoptosis, is induced by p53. P53 increases the expression of p21 and p16, which in turn causes cell cycle arrest to induce senescence. It is probable that low doses of chemotherapy drugs cause senescence via p53 while higher doses of chemotherapy drugs cause apoptosis via p53.
Please mark the explanation in the manuscript, I could not find it.
Please add some references, especially where you found that p53 increases the expression of p16.
Q6.Figure 2: this figure is very busy and therefore difficult to understand. I suggest that the authors separate the figure into a figure that shows the stimuli and mechanisms of senescence induction and the therapy-induced senescence induction.
Ans. I agree. I separate figure 2 into two parts. Please take look at new figure 2.
Chapter “Oncogene-induced Senescence”
Q1. Line 97: As the authors describe that anti-cancer therapies like chemotherapeutic drugs or radiotherapies induce DNA damage and ROS and do not target oncogenes, this sentence does not make sense. Please remove it.
Ans. I agree. I remove it. Please take look at new manuscript.
Q2.Line 100: Please explain the contradictory findings that senescent cancer cells, although permanently growth arrested as described in line 22, become more aggressive.
Ans. Oncogenes can promote senescence by inducing DNA damage and ROS production. Not all senescent cells remain permanently arrested. Some senescent cells remain active and secrete inflammatory cytokines/chemokines for communications with nearby cells. These cellular communications may allow senescent cells to acquire aggressive phenotypes.
Line 117: “some senescent cells remain active and secrete inflammatory cytokines/chemokines for communications with nearby cells”.
Most, if not all senescent cells, independent of permanent cell cycle arrest or not, secrete inflammatory cytokines known as the senescence-associated secretory phenotype (SASP). Senescent cells that escape the stable growth arrest harbour epigenetic and/or genetic mutations and not pro-inflammatory SASP factors.
I think, what the authors mean is that the SASP can promote the malignant conversion of otherwise non-malignant cells. However, this is not the escape of senescent cells from growth arrest.
I recommend that the authors read the review of Clemens Schmitt, Boshi Wang and Marco Demaria in the October issue of Nature Reviews Clinical Oncology.
Q3. Line 102: immune checkpoint inhibitor therapy tries to activate immune cells. Why do cancer cells like TNBCs react or even get resistant after antibody injection? Please explain.
Ans. Immune check point inhibitor (ICI), such as anti-PD-L1, can bind to PD-L1 expressed on TNBCs. This binding suppresses the binding of PD-L1 to PD-1 to inhibit immune evasion exerted by cancer cells. Primary resistance mechanisms (to ICI) include insufficient tumor immunogenicity, dysfunction of MHCs, irreversible T cell exhaustion, primary resistance to IFN-γ signaling, and immunosuppressive microenvironment. Some oncogenic signaling pathways also contribute to the primary resistance. Under the pressure applied by anti-PD1/PDL1 therapy, tumors experience immunoediting and preserve beneficial mutations, upregulate the compensatory inhibitory signaling and induce re-exhaustion of T cells, all of which may attenuate the durability of the therapy.
Q4. Line 105: I do not understand these sentences. The authors write that targeting oncogenes leads to the elimination of senescent cells. Like treatment with senolytic agents? In addition, why should this prevent transformation of senescent cells? Please explain this.
Ans. It is known that senolytic drugs target senescent cancer cells. Oncogenes can induce senescence. Some transformed senescent cells can remain active and acquire aggressive phenotypes by secreting inflammatory cytokines/chemokines to induce cellular interactions. Therefore, targeting oncogenes can eliminate these senescent cancer cells by preventing them from inducing cellular interactions for acquiring malignant phenotypes.
I still do not understand this explanation. If you target oncogenes like RAS, you prevent oncogene-induced senescence. A non-senescent cell does not have a SASP that can promote malignant transformation of otherwise non-malignant cells. In the paper you cite (48) do the authors show that targeting MYC in the senescent TNBCs reverts these cells into non-senescent cells that have no SASP anymore?
Q5. The authors show that most OIS induction is p53 dependent. In human cancer cells TP53 and RB1 pathways are frequently mutated although, most cancer cell lines can be rendered senescent in vitro by e.g., AURK inhibitors. Is there an explanation for this?
Ans. It has been known that aurora kinase inhibitors induce the expression of p53 and/or senescence:
- Liu Y, Hawkins OE, Su Y, Vilgelm AE, Sobolik T, Thu YM, Kantrow S, Splittgerber RC, Short S, Amiri KI, Ecsedy JA, Sosman JA, Kelley MC, Richmond A. Targeting aurora kinases limits tumour growth through DNA damage-mediated senescence and blockade of NF-κB impairs this drug-induced senescence. EMBO Mol Med. 2013 Jan;5(1):149-66. doi: 10.1002/emmm.201201378.
- Liu X, Shi Q, Choudhry N, Zhang T, Liu H, Zhang S, Zhang J, Yang D. The Effect of Circumscribed Exposure to the Pan-Aurora Kinase Inhibitor VX-680 on Proliferating Euploid Cells. Int J Mol Sci. 2022 Oct 11;23(20):12104. doi: 10.3390/ijms232012104.
Since AURK inhibitors induce the expression of p53, these inhibitors can induce senescence.
Of course, AURK inhibitors can induce the expression of p53. Nevertheless, in tumor cells that harbour a mutated p53 the inhibitors cannot induce wildtype p53 because it is mutated. Still, many tumor cells with a mutated p53 can undergo senescence in vitro by AURK inhibitors. How? Is there an explanation for that?
Q6. Line 153: The authors write in line 100 that senescent cells are permanently growth arrested. In line 145, the authors write that senescent cells transform into cancer cells. Is a cell that will never divide a cancer cell anymore?
Ans. Senescent cells are unique in that they eventually stop multiplying but don’t die off when they should. Not all senescent cells remain permanently arrested. Some senescent cells remain active and secrete inflammatory cytokines/chemokines for communications with nearby cells. These cellular interactions enable senescent cells to acquire aggressive phenotypes.
This is the same problem as before. Most senescent cells, if not all senescent cells are metabolically very active and secrete pro-inflammatory factors, the SASP. Although they are cell arrested. The escape from senescence or the acquisition of a stem cell phenotype is, as far as I know, not a function of inflammatory factors but of epigenetic and genetic mutations in the senescent cell.
Q7.To avoid confusion about the point that a senescent cell is permanently growth arrested and can transform into a cancer cell the authors should add a paragraph about the stability of senescence and how senescence is maintained.
Ans. Thanks. I agree. I add this sentence: Not all senescent cells remain permanently arrested. Some senescent cells remain active and secrete inflammatory cytokines/chemokines for communications with nearby cells. These cellular communications may allow senescent cells to acquire aggressive phenotypes. Please take look at lines 124-127.
Chapter “Roles of SASP in Cancer Cell Proliferation and Therapeutic Resistance”
Q1. The authors should explain that the SASP differs according to the senescence inducing stimuli. Judith Campisi and collaborators were the first that described the different SASP factors 2016.
Ans. Thanks. I add this sentence: Composition of SASP shows variation depending on the stimuli, the cell type, and the stage of senescence. Please take look at lines 671-672. I add references below.
Lines 671-672 are a small part of the reference list.
â‘ Wiley CD, Velarde MC, Lecot P, Liu S, Sarnoski EA, Freund A, Shirakawa K, Lim HW, Davis SS, Ramanathan A, et al. Mitochondrial Dysfunction Induces Senescence with a Distinct Secretory Phenotype. Cell Metab. 2016 23,303-14. doi: 10.1016/j.cmet.2015.11.011.
â‘¡Wiley CD, Flynn JM, Morrissey C, Lebofsky R, Shuga J, Dong X, Unger MA, Vijg J, Melov S, Campisi J. Analysis of individual cells identifies cell-to-cell variability following induction of cellular senescence. Aging Cell 2017, 16, 1043-1050. doi: 10.1111/acel.12632.
Q2. Line 172: reference 57 is a study in zebrafish and not in humans.
Ans. I change. Thanks. Please take look at line 201.
Q3. Line 178: what kind of cells form this tumor-like cell mass? Please explain.
Ans. Tumor-like cell mass include transformed RASG12V mutant cells that became senescent and eliminated, RASG12V-TP53R175H double-mutant cells, and normal cells. The surviving double mutant cells secrete SASP-like inflammatory cytokines to convert nearby normal cells into SASP factor-secreting senescent cells.
Q4. Line 216: the authors explain the resistance of senescent cells to anticancer drugs with the upregulation of anti-apoptotic proteins. Most chemotherapeutic drugs need highly proliferating cells to be effective and senescent cells are growth arrested, this could be another explanation for the resistance. Please discuss other explanations.
Ans. Senescence plays paradoxical roles in cancer. Induction of senescence inhibits cancer progression and enhances the sensitivity to anticancer drug but lingering (metabolically active) senescent cells fuel progression, recurrence, and metastasis. Lingering senescent cells are metabolically active and can display resistance to anticancer drugs. These lingering senescent cells secrete SASP to affect nearby cells. Cellular interactions driven by SASP may affect phenotypes of these senescent cells.
Q5. Line 215: With this very interesting paper, it is the same problem as before. Permanently arrested cells became proliferating cancer cells. I recommend that the authors insert a paragraph that describes the stability of senescence.
Ans. I agree. I add these sentences: Senescent cells are unique in that they eventually stop multiplying but don’t die off when they should. A relatively small number of senescent cells can remain active and spread senescence (lines 39-41). The irreversible senescence is mediated by INK4a/Rb and p53 pathways and the reversible senescent phenotype is mediated by p53. This suggests that p53 pathway could be effectively harnessed as a therapeutic intervention to trigger senescence and ultimately suppress tumorigenesis (lines 71-74). Senescent cells can undergo conversion to an immunogenic phenotype that enables them to be eliminated by the immune system (lines 327-328). I hope that these sentences may give partial answers.
Lines 39-41: How do you know that only a small number of senescent cells are metabolically active and secrete SASP factors while most of the senescent cells are inactive without a SASP? If this is true, please add a reference supporting this statement.
Q6. Line 220: targeting BCL-2 or other anti-apoptotic proteins is known as senolysis or senolytic therapy.
Ans. Thanks. I agree. I add this sentence: Thus, targeting BCL-2 or other anti-apoptotic proteins can eliminate senescent cancer cells (senolysis or senolytic therapy). Please take look at lines 254-256.
Q7. Line 212: I have never heard that EGFR and HER2 tyrosine kinases inhibitors like lepatinib and fulvestran can induce senescence. Please explain this astonishing effect.
Ans. Thanks. I agree. I change the sentence into: Lapatinib and fulvestrant (EGFR and HER2 inhibitors) treatment induced apoptosis in chemotherapy-induced senescent breast cancer cells [83]. Please take look at lines 251-252.
Q8. Line 224: inflammatory endothelial cells secrete high levels of IL-8 without being senescent. Targeting IL-8 can of course improve clinical outcome but not necessary as part of the SASP.
Ans. Thanks. I agree. I thank your outstanding perspectives. I just wanted to say some possible role of IL-8 in anticancer drug resistance. IL-8 can be a component of SASP in a context-dependent manner.
Chapter “Targeting Therapy-induced senescence”
- In my opinion, this paragraph is not necessary because targeting TIS is already described in the previous paragraphs and the MDM2-p53 disruptors like nutlin 3 can be added in the “Factors inducing Senescence” paragraph.
Ans. Thanks. I agree. I delete the paragraph. Please take look at new manuscript. I put reference concerning MDM2-p53 disruptors like nutlin 3 into “Factors inducing Senescence” paragraph. Reference 124 is now ref 24.
Chapter “Role of Autophagy in Senescence”
Q1. Line 359: the authors write that it is still unclear whether autophagy has a positive or negative effect on senescence but in the whole paragraph they only cite references that show a positive effect of autophagy on cancer cell senescence. Please add examples for negative effects of autophagy on senescence.
Ans. I add this reference: Fan X, He Y, Wu G, Chen H, Cheng X, Zhan Y, An C, Chen T, Wang X. Sirt3 activates autophagy to prevent DOX-induced senescence by inactivating PI3K/AKT/mTOR pathway in A549 cells. Biochim Biophys Acta Mol Cell Res. 2023,1870, 119411. doi: 10.1016/j.bbamcr.2022.119411. This shows a negative regulatory role of autophagy in senescence.
Please take look at lines 360-361 and new reference (ref. 114).
Q2.Line 375: just because both autophagy and senescence induce anticancer drug resistance, one cannot say that CAGE induces senescence. To relate autophagy and senescence with the reference the authors cite is very speculative.
Ans. I agree. I rather delete this sentence: Since senescence can induce anticancer drug resistance, it is probable that CAGE may also induce senescence to promote anticancer drug resistance. Please see new manuscript.
Q3. Line 408: is this sentence true for all anti-cancer drugs and all senescence-inducing stimuli?
Ans. I think that it may not be true for all anti-cancer drugs. I rather delete the sentence: Thus, targeting autophagy may enhance the effects of anticancer drugs by suppressing senescence. Please take look at new manuscript.
Chapter “Role of HDACs in Senescence”
Q1. This paragraph is very inconsistent, it describes HDACs and apoptosis and senescence and histone methylation. I recommend to rewrite it and only describe HDACs and senescence.
Ans. I agree. I remove unnecessary sentences and references. I try to mention only HDACs and senescence. Please take look at new manuscript. New paragraph focuses on HDACs and senescence.
Q2. Line 476: please explain what HDACs are and why they are important epigenetic regulators.
Ans. It is well known that histone modifications regulate gene expression. Tet methylcytosine dioxygenase 2 (TET2) inhibited PD-L1 gene expression by recruiting HDACs in MDA-MB-231 cells [136].
They are HDAC1 and -2. TET2 recruits HDAC1/2 to deacetylate H3K27ac at PD-L1 promoter. Histone deacetylation at PD-L1 promoter decreases the expression of PD-L1. It is known that PD-L1/PD-1 axis is necessary for senescence. Thus, immune checkpoint inhibitor (ICI) therapy can promote senescence surveillance which eliminates senescent cancer cells. The reference below shows the effect of ICI on senescence surveillance. It is thus reasonable that HDAC1/2 may induce immune surveillance (senescence surveillance) by decreasing the expression of PD-L1.
Wang TW, Johmura Y, Suzuki N, Omori S, Migita T, Yamaguchi K, Hatakeyama S, Yamazaki S, Shimizu E, Imoto S, Furukawa Y, Yoshimura A, Nakanishi M. Blocking PD-L1-PD-1 improves senescence surveillance and ageing phenotypes. Nature. 2022 Nov;611(7935):358-364. doi: 10.1038/s41586-022-05388-4.
Thank you very much for this explanation but I am sure that some readers of this review do not know what HDACs are. You not even explain the abbreviation HDAC. At least I could not find that HDACs are histone deacetylases.
Q3. Line 478: the authors describe the methylation of histones in the HDAC paragraph. Please do not mingle acetylation and methylation.
Ans. I agree. I understand. I just wanted to show that histone methylation, just like histone acetylation/deacetylation, can affect senescence. In other words, histone modifications generally could affect senescence. I do not intend to mingle acetylation and methylation. I am sorry to cause confusion.
Q4. Line 484: not every cell that secretes a pro-inflammatory cytokine like IL-8 is senescent. Please specify the markers for senescence used in this study.
Ans. This study does not show the effect of 4-Phenylbutyric acid (4-PBA), an HDAC inhibitor (HDACi) on senescence. However, 4-PBA decreased the expression of IL-8, a component of SASP. I just wanted to show a possible role of HDACs in senescence. I agree that this study does not show direct relationship between HDACs and senescence. I am sorry to cause confusion.
Q5. Line 494: caspase 3/7 is no senescence marker but a marker for apoptosis. The inhibition of HDAC2/HDAC6 together with cisplatin seems to enhance apoptosis and not senescence.
Ans. I agree. I delete the sentence and the reference. Please take look at new manuscript.
Q6.Line 496: the authors want to discuss the role of HDACs in senescence and not the role of HDACs in apoptosis.
Ans. I agree. I delete the sentence and the reference. Please take look at mew manuscript.
Chapter “Conclusion and Perspective”
Q1. Line 574-579: In the first sentence, the authors say that senescent cells can acquire an aggressive phenotype and in the second sentence, the senescent tumour cell is permanently growth arrested. This contradiction must be explained.
Ans. I agree. It is my mistake. Senescent cells are unique in that they eventually stop multiplying, but not die off. I did not mean that all senescent cells are permanently growth arrested. All senescent cells do not stay permanently arrested. Some senescent cells remain metabolically active and secrete pro-inflammatory cytokines. A small number of senescent cells remain and release chemicals that can cause inflammation in nearby cells. These cellular interactions enable senescent cells to acquire aggressive phenotypes. Please take look at lines 39-41.
Q2. Line 581: How does CopA3 induce senescence? Like a CDK4/6 inhibitor?
Ans. Defensin-like peptide CopA3 increased SA-β-gal activity and ROS levels and DNA damage and G1 cell cycle arrest in colon cancer cells. It is not clear whether this peptide display CDK inhibition effect. Since it induces G1 arrest and senescence, it is possible that CopA3 peptide can decrease CDK activity. It is well known that CDK is necessary for cell cycle progression.
Q3.Line 585-585: in the first sentence the authors explain that factors of the SASP of senescent cells can induce the upregulation of PD-L1. In the third sentence the authors write that upregulation of PD-L1 on non-senescent cells by chemotherapeutic drugs induces senescence. Is the upregulation of a checkpoint inhibitor molecule enough to induce senescence? In the previous paragraphs the authors explain that chemotherapeutic drugs induce senescence by DNA damage. Please explain these contradictory findings.
Ans. The references below show that doxorubicin increases the expression of PD-L1. The increased expression of PD-L1 causes immune evasion. PD-L1/PD-1 axis has been known to be involved in senescence. The blocking of PD-L1/PD-1 axis induces senescence surveillance (elimination of senescent cells). I would not say upregulation of PD-L1 alone can induce senescence. However, upregulation of PD-L1, at least partially, is necessary for the induction of senescence by chemotherapy. Thus, chemotherapy drugs not only induce DNA damage but also induce PD-L1 expression.
â‘ Wang J, Hu F, Yu P, Wang J, Liu Z, Bao Q, Zhang W, Wen J.J. Sorafenib inhibits doxorubicin-induced PD-L1 upregulation to improve immunosuppressive microenvironment in Osteosarcoma. Cancer Res Clin Oncol. 2022 Nov 8. doi: 10.1007/s00432-022-04458-4. Online ahead of print.
â‘¡Yati S, Silathapanasakul A, Thakaeng C, Chanasakulniyom M, Songtawee N, Porntadavity S, Pothacharoen P, Pruksakorn D, Kongtawelert P, Yenchitsomanus PT, Chanmee T. Extracellular Vesicle-Mediated IL-1 Signaling in Response to Doxorubicin Activates PD-L1 Expression in Osteosarcoma Models. Cells. 2022 Mar 18;11(6):1042. doi: 10.3390/cells11061042.
â‘¢Wang TW, Johmura Y, Suzuki N, Omori S, Migita T, Yamaguchi K, Hatakeyama S, Yamazaki S, Shimizu E, Imoto S, Furukawa Y, Yoshimura A, Nakanishi M. Blocking PD-L1-PD-1 improves senescence surveillance and ageing phenotypes. Nature. 2022 Nov;611(7935):358-364. doi: 10.1038/s41586-022-05388-4.
Q4. Line 590: I do not understand this sentence. More PD-L1 on senescent cancer cells should enhance the efficiency of PD-1 antibodies that target PD-1 on immune cells? How and why?
Ans. For example, anti-PD-L1 antibody can bind to PD-L1 on senescent cancer cells. The binding of anti-PD-L1 antibody to PD-L1 inhibits the binding of PD-L1 to PD-1 to suppress immune evasion. Therefore, the increased expression of PD-L1 on senescent cancer cells enhances the sensitivity of cancer cells to PD-L1/PD-1 blockade.
Q5. Line 605: the main task of senolytic drugs is the elimination of senescence cells by targeting anti-apoptotic pathways!
Ans. I add the sentence: the main task of senolytic drugs is the elimination of senescence cells by targeting anti-apoptotic pathways. Please take look at lines 614-615.
Please also take look at lines 620-622:BCL2-targeting drugs, including ABT-737 and ABT-263 (navitoclax), natural substances, such as artesunate, fisetin, and curcumin, showed senolytic effects in glioblastoma cells and non-transformed cells [82,166].
Q6.Line 609: usually cells upregulate more than one ant-apoptotic proteins so targeting only the BCL-pathway should result in apoptosis induction in a restricted range of senescent cell types.
Ans. Thanks. I agree. I add this reference. Please take look at lines. 617-619. This reference shows Wnt/β-catenin signaling as another target of senolytic drugs.
Qin G, Li Y, Xu X, Wang X, Zhang K, Tang Y, Qiu H, Shi D, Zhang C, Long Q, et al. Panobinostat (LBH589) inhibits Wnt/β-catenin signaling pathway via upregulating APCL expression in breast cancer. Cell Signal. 2019 Jul;59:62-75. doi: 10.1016/j.cellsig.2019.03.014.
Q7.Line 640: despite the Silybum marianum flower extract the authors mention there are more SASP inhibitors like Rapamycin or the anti-diabetic drug metformin. Please discuss some more SASP inhibitors.
Ans. I add reference below (rapamycin). I also add this sentence: Rapmycin, an inducer of autophagy, inhibits senescence and SASP by decreasing SA-β-Gal activity and p16 expression in human coronary artery endothelial cells (ref 174). Please take look at lines 664-666.
Sasaki N, Itakura Y, Toyoda M. Rapamycin promotes endothelial-mesenchymal transition during stress-induced premature senescence through the activation of autophagy.
Cell Commun Signal. 2020 Mar 12;18(1):43. doi: 10.1186/s12964-020-00533-w.
Q8. Line 649: please specify the peptide.
Ans. Thanks. It is defensin-like CopA3 peptide. It is already mentioned. I rather delete the sentence.
Q9. Please describe the advantages and disadvantages of senolytics versus SASP inhibitors.
I mentioned them in the manuscript. Please take look at lines (640-645) and (667-672).
Ans.
< Senolytics >
Senescent cells are like cancer cells that do not divide, including metabolic shift, epigenetic change, and resistance to apoptosis. Senescent cells increase several networks of anti-apoptotic regulators, PI3K/Akt pathway components, and BCL-2 family members, which collectively confer resistance to apoptosis. First generation of senolytics inhibit a portion of these pathways and induce apoptosis preferentially in senescent cells. There are two critical considerations for developing and applying senolytics: first, it is necessary to target a whole pathway for survival of senescent cells, not a single gene in that pathway as other components in the same pathway can compensate for its loss. Second, aiming for one survival pathway alone is not sufficient to eliminate all different populations of senescent cells as they are quite heterogenous.
The current version of senolytics has its own limitation. Since most senolytic drugs were repositioned to target aging and age-related diseases, they might have undesirable side effects for long-term use. Therefore, it is required to reevaluate the safety of a senolytic drug for long-term use if it is originally designed for short-term use. Another potential problem is tissue atrophy, resulting from a massive removal of senescent cells by senolytic drugs. Senescent cells play a role to support the structure of old tissues and thus their abrupt removal may lead to atrophy, depending on the levels of senescent cells that reside in the tissue (He and Sharpless, 2017). In addition, senolytics wipes out not only deleterious effects of senescent cells but also their beneficial effects; cellular senescence contributes to wound healing (Demaria et al., 2014), cellular reprogramming (Mosteiro et al., 2016), and tissue regeneration (Ritschka et al., 2017), suggesting that complete elimination of senescent cells may hamper these beneficial effects of senescent cells.
<SASP inhibitors>
For generalized senostatics, several theraputic targets have been recently emerged as the SASP regulatory network: NF-κB, p38, GATA4, mTOR, BRD4, and cGAS/STING. One potential problem for targeting such regulatory network is that they have also non-senescence related functions; NF-κB plays an essential role in controlling acute inflammatory response and immune response (Chien et al., 2011); p38 is involved in several stress responses including DNA damage, heat shock, and osmotic shock responses (Freund et al., 2011); GATA4 plays a major role in controlling embryonic development (Molkentin et al., 1997); mTOR governs cell growth, proliferation, protein synthesis, and autophagy (Herranz et al., 2015; Laberge et al., 2015); BRD4 functions as an epigenetic reader to modulate lineage- and cellular state-specific transcription (Tasdemir et al., 2016); finally, cGAS-STING is a component of the innate immune system that senses the presence of cytosolic DNA (Yang et al., 2017). Therefore, care must be considered to tailor therapeutic strategies to exclusively blunt the SASP regulation but no other functions. Precision senostatics have not been seriously considered yet mainly because the composition of the SASP highly varies, depending on the cell type, the stage of senescence (early versus late senescence), and the type of senescence-inducing stimuli. Moreover, the composition of the SASP is quite complex, having up to a slightly less than hundred factors. Future studies need to categorize SASP factors according to their functions in a context-dependent manner.
Author Response
Dear Sir
Thanks for excellent suggestions. I learned a lot from this process. I made changes according to the suggestions made by reviewer. I hope that I made are fine.
Sincerely yours
Jeoung Dooil
Q. Subterranean wild rodents are quite an exotic animal model. Is there no study in human cancer cell lines or human specimens? At least Etoposide is a chemotherapeutic drug applied in the clinic.
- Ans. I add this sentence: Etoposide, an inhibitor of topoisomerase, induced senescence by increasing the expression of p53 and p21 via DNA protein kinase (DNA-PK)-Checkpoint kinase 2 (Chk2 pathway) in adrenocortical carcinoma cells [23]. Please take look lines 61-63.
I add new reference below. I delete old reference.
Teng, Y.N.; Chang, H.C.; Chao, Y.Y.; Cheng, H.L.; Lien, W.C.; Wang, C.Y. Etoposide Triggers Cellular Senescence by Inducing Multiple Centrosomes and Primary Cilia in Adrenocortical Tumor Cells. Cells 2021, 10, 1466. doi: 10.3390/cells10061466
Q. My question was about platinum-based drugs and not topoisomerase inhibitors! Please write one or two sentences to platinum-based chemotherapeutic drugs.
- Ans. Thanks. I agree. I delete old sentence. I add this sentence: Platinum-based drug, such as cisplatin, induced senescence by up-regulation of senescence-related genes (p53, p21 and p16) in seminoma cells [26]. Cisplatin induced DNA damage and increased ROS level [26]. Please take look at lines 73-75. I also add the reference below.
Yin, Y.; Peng, J.; Zheng, X.; Zhou, J.; Wang, Y.; Dai, Y.; Yin, G.; Tang, Y. Extrinsic apoptosis and senescence involved in growth kinetics of seminoma to cisplatin. Clin Exp Pharmacol Physiol 2023, 50,140-148. doi: 10.1111/1440-1681.13730
Q. Please explain why chemotherapies and radiotherapies that induce DNA damage and subsequently p53 upregulation induce senescence although p53 is known as regulator of apoptosis.
Please mark the explanation in the manuscript, I could not find it.
Please add some references, especially where you found that p53 increases the expression of p16.
- Ans.P53 displays diverse roles, including apoptosis and senescence. P53 activation in response to DNA damage causes growth arrest, allowing for DNA repair, or directs senescence or apoptosis [25]. For DNA repair, senescence, but not apoptosis, is induced by p53. P53 increases the expression of p21 and p16, which in turn causes cell cycle arrest to induce senescence [25]. It is probable that low doses of chemotherapy drugs cause senescence via p53 while higher doses of chemotherapy drugs cause apoptosis via p53. Please take look at lines 66-73. I add reference below.
Eleftheriadis, T.; Pissas, G.; Filippidis, G.; Efthymiadi, M.; Liakopoulos, V.; Stefanidis, I. Dapagliflozin Prevents High-Glucose-Induced Cellular Senescence in Renal Tubular Epithelial Cells.
Int J Mol Sci 2022, 23,16107. doi: 10.3390/ijms232416107
Q. Line 117: “some senescent cells remain active and secrete inflammatory cytokines/chemokines for communications with nearby cells”.
Most, if not all senescent cells, independent of permanent cell cycle arrest or not, secrete inflammatory cytokines known as the senescence-associated secretory phenotype (SASP). Senescent cells that escape the stable growth arrest harbour epigenetic and/or genetic mutations and not pro-inflammatory SASP factors.
I think, what the authors mean is that the SASP can promote the malignant conversion of otherwise non-malignant cells. However, this is not the escape of senescent cells from growth arrest.
I recommend that the authors read the review of Clemens Schmitt, Boshi Wang and Marco Demaria in the October issue of Nature Reviews Clinical Oncology.
- Ans. Thanks. I agree. I read the review. I add these sentences: Continuous functional changes in senescent cells, and the permanent need to actively maintain senescence-supporting transcription puts the stability of cell cycle arrest at risk. Oncogene induced senescence (OIS) can induce permanent growth arrest. OIS can also enable senescent cells to evade the potential toxicity of therapeutics, allowing for the eventual re-emergence or escape from senescence that could lead to disease recurrence. senescence is potentially reversible through the inactivation of p53, p16(INK4A) and/or Rb, over-expression of Cdc2/cdk1, and the survival of cancer stem cells. Please take look at lines 126-133.
I delete these sentences: Not all senescent cells remain permanently arrested. Some senescent cells remain active and secrete inflammatory cytokines/chemokines for communications with nearby cells. These cellular communications may allow senescent cells to acquire aggressive phenotypes
Q. I still do not understand this explanation. If you target oncogenes like RAS, you prevent oncogene-induced senescence. A non-senescent cell does not have a SASP that can promote malignant transformation of otherwise non-malignant cells. In the paper you cite (48) do the authors show that targeting MYC in the senescent TNBCs reverts these cells into non-senescent cells that have no SASP anymore?
- Ans. I agree. I am sorry to cause confusion. I rather delete the sentence: Thus, targeting oncogenes can eliminate senescent cells, preventing the transformation of senescent cells into cancer cells. Senolytic drugs eliminates senescent cells that can overcome permanent arrest
Q. Of course, AURK inhibitors can induce the expression of p53. Nevertheless, in tumor cells that harbour a mutated p53 the inhibitors cannot induce wildtype p53 because it is mutated. Still, many tumor cells with a mutated p53 can undergo senescence in vitro by AURK inhibitors. How? Is there an explanation for that?
- Ans. Thanks. I agree. I add new reference below. This reference shows the induction of senescence in p53-independent pathway.
Liu, Y.; Hawkins, O.E.; Su, Y.; Vilgelm, A.E.; Sobolik, T.; Thu, Y.M.; Kantrow, S.; Splittgerber, R.C.; Short, S.; Amiri, K.I. et al. Targeting aurora kinases limits tumour growth through DNA damage-mediated senescence and blockade of NF-κB impairs this drug-induced senescence. EMBO Mol Med 2013, 5,149-166. doi: 10.1002/emmm.201201378.
I add this sentence: The inhibition of aurora kinases induced polyploidy and the ATM/Chk2 DNA damage response, which mediated senescence and a NF-κB-related, senescence-associated secretory phenotype (SASP) [32]. This implies that AURK inhibitors can also induce senescence in p53-indepndent way. Please take look at lines 92-96.
Q. This is the same problem as before. Most senescent cells, if not all senescent cells are metabolically very active and secrete pro-inflammatory factors, the SASP. Although they are cell arrested. The escape from senescence or the acquisition of a stem cell phenotype is, as far as I know, not a function of inflammatory factors but of epigenetic and genetic mutations in the senescent cell.
- Ans. Thanks. I agree. It is rather epigenetic and genetic mutations in the senescent cells that overcomes permanent arrest.
I add these sentence: Continuous functional changes in senescent cells, and the permanent need to actively maintain senescence-supporting transcription puts the stability of cell cycle arrest at risk. Oncogene induced senescence (OIS) can induce permanent growth arrest. OIS can also enable senescent cells to evade the potential toxicity of therapeutics, allowing for the eventual re-emergence or escape from senescence that could lead to disease recurrence. Senescence is potentially reversible through the inactivation of p53, p16(INK4A) and/or Rb, over-expression of Cdc2/cdk1, and the survival of cancer stem cells. Please take look at lines 126-133.
Q. Lines 671-672 are a small part of the reference list.
- Ans. Thanks. I made changes. I add these sentences: Analysis of senescent human fibroblast cells revealed that individual senescent cells showed heterogeneity in their gene expression signatures [177]. This heterogeneity led to a weak correlation among genes encoding SASP [177]. Composition of SASP shows variation depending on the stimuli, the cell type, and the stage of senescence [177]. Please take look at lines 642-644.
I add reference below:
Wiley CD, Flynn JM, Morrissey C, Lebofsky R, Shuga J, Dong X, Unger MA, Vijg J, Melov S, Campisi J. Analysis of individual cells identifies cell-to-cell variability following induction of cellular senescence. Aging Cell 2017, 16, 1043-1050. doi: 10.1111/acel.12632.
Q. Lines 39-41: How do you know that only a small number of senescent cells are metabolically active and secrete SASP factors while most of the senescent cells are inactive without a SASP? If this is true, please add a reference supporting this statement.
- Ans. Thanks. I agree. I change the sentence into: Senescent cells can remain active and spread senescence. I delete “only a small number of “
Q. Thank you very much for this explanation but I am sure that some readers of this review do not know what HDACs are. You not even explain the abbreviation HDAC. At least I could not find that HDACs are histone deacetylases.
- Ans. HDACs are histone deacetylases. Please take look at line 468. I thought that I did not have to mention that HDACs play important roles in various life processes.
